# GeLoc3r: Enhancing Relative Camera Pose Regression with Geometric Consistency Regularization

## Abstract

Prior ReLoc3R achieves breakthrough performance with fast 25ms inference and state-of-the-art regression accuracy, yet our analysis reveals subtle geometric inconsistencies in its internal representations that prevent reaching the precision ceiling of correspondence-based methods like MASt3R (which require 300ms per pair). In this work, we present GeLoc3r, a novel approach to relative camera pose estimation that enhances pose regression methods through Geometric Consistency Regularization (GCR). GeLoc3r overcomes the speed-accuracy dilemma by training regression networks to produce geometrically consistent poses without inference-time geometric computation. During training, GeLoc3r leverages ground-truth depth to generate dense 3D-2D correspondences, weights them using a FusionTransformer that learns correspondence importance, and computes geometrically-consistent poses via weighted RANSAC. This creates a consistency loss that transfers geometric knowledge into the regression network. Unlike FAR method which requires both regression and geometric solving at inference, GeLoc3r only uses the enhanced regression head at test time, maintaining ReLoc3R's fast speed and approaching MASt3R's high accuracy. On challenging benchmarks, GeLoc3r consistently outperforms ReLoc3R, achieving significant improvements including 40.45% vs. 34.85% AUC@5° on the CO3Dv2 dataset (16% relative improvement), 68.66% vs. 66.70% AUC@5° on RealEstate10K, and 50.45% vs. 49.60% on MegaDepth1500. By teaching geometric consistency during training rather than enforcing it at inference, GeLoc3r represents a paradigm shift in how neural networks learn camera geometry, achieving both the speed of regression and the geometric understanding of correspondence methods.

## 1 Introduction

Estimating relative camera poses between image pairs is fundamental in 3D vision applications such as structure-from-motion (SfM) (Schönberger & Frahm, 2016), visual SLAM (Mur-Artal et al., 2015), and augmented reality. The field has traditionally faced a speed-accuracy dilemma where correspondence-based methods that explicitly match features and solve geometric constraints achieve high accuracy but require 300-400ms per pair (Leroy et al., 2024; 2023), while early regression-based methods offered fast inference but with significantly lower accuracy. Recent ReLoc3R (Dong, 2024) achieves a breakthrough by leveraging Vision Transformers with cross-attention mechanisms and large-scale training to deliver both fast inference (25ms) and competitive accuracy, substantially closing the historical performance gap. However, despite this impressive advance, a persistent accuracy ceiling remains—ReLoc3R and other regression methods still cannot fully match the precision of correspondence-based approaches. FAR (Rockwell & Fouhey, 2024) partially addressed this by demonstrating complementary strengths and fusing both paradigms at inference, but this requires running both methods, negating the speed advantage.

Our analysis reveals the root cause of this accuracy ceiling. While ReLoc3R produces reasonable pose predictions, its internal descriptor representations lack geometric consistency—the network learns rich semantic features but fails to maintain the correspondence relationships essential for precise pose estimation. This fundamental disconnect between semantic understanding and geometric accuracy creates an inherent limitation that architectural improvements alone cannot overcome. The

challenge lies in a fundamental barrier that has prevented prior works from incorporating geometric constraints into regression training: geometric constraints traditionally require depth information unavailable at inference time.

In this work, we propose GeLoc3r, which introduces Geometric Consistency Regularization (GCR)—a novel training paradigm that teaches regression networks to understand 3D geometry without requiring geometric computation at inference. By leveraging ground-truth depth available in training datasets to provide geometric supervision, our method maintains the 33ms inference speed of pure regression while approaching the accuracy of correspondence-based methods.

GeLoc3r achieves state-of-the-art regression results across six benchmarks. Most notably, on the CO3Dv2 dataset, we achieve 40.45% AUC@5° compared to ReLoc3R's 34.85%. We also consistently outperform ReLoc3R on other datasets, improving AUC@5° from 66.70% to 68.66% on RealEstate10K and achieving 82.55% AUC@20° on MegaDepth1500 with extreme viewpoint changes (vs. 81.2% for ReLoc3R). Furthermore, GeLoc3r demonstrates strong generalization on entirely unseen visual localization tasks, achieving 0.04m/1.09° on indoor 7-Scenes and 0.51m/0.52° on outdoor Cambridge Landmarks. These consistent improvements validate that incorporating geometric consistency during training fundamentally enhances regression networks without compromising inference efficiency.

## 2 RELATED WORK

Pose Regression (PR) methods directly predict camera poses through end-to-end neural networks without explicit geometric solving.

**Learning-based Pose Estimation.** Recent works have explored end-to-end learning for relative pose estimation. RelPoseNet (Zhang et al., 2022) uses CNNs for direct regression, while ReLoc3R (Dong, 2024) leverages Vision Transformers with cross-attention for improved geometric reasoning. Chen et al. (Chen et al., 2021) propose DirectionNet which decomposes relative pose into 3D direction vectors on the sphere, achieving significant improvements for wide-baseline scenarios. Xu et al. (Xu et al., 2023) integrate rotation-consistent global features with rotation-invariant multi-level local features to improve pose regression robustness. These architectural and feature design improvements have advanced regression performance. However, regression-based approaches face well-documented limitations. Sattler et al. (Sattler et al., 2019) demonstrated that pose regression methods systematically achieve lower accuracy than structure-based methods, showing they behave more like image retrieval than geometric pose estimation. The core challenge, as identified by Kendall and Cipolla (Kendall & Cipolla, 2017), stems from the difficulty of balancing rotation and translation components that have fundamentally different scales and units (degrees vs. meters), requiring either manual hyperparameter tuning or learned uncertainty weighting. Our work addresses these limitations from a different angle—instead of improving network architecture or features, we introduce training-time geometric consistency regularization that teaches regression networks to produce geometrically consistent poses without requiring architectural changes.

**Dense Correspondence Methods.** DUSt3R (Leroy et al., 2023) and MASt3R (Leroy et al., 2024) predict dense 3D points and descriptors, enabling robust matching through confidence-aware sampling. These approaches excel when sufficient correspondences exist but require additional solver steps. DUSt3R suffers from slow inference speed due to the high computational complexity of dense reciprocal matching, which is $O(W^2H^2)$ (Leroy et al., 2023). MASt3R mitigates this with fast nearest neighbor search, reducing complexity to $O(kWH)$ and accelerating matching by up to 64 times while improving performance (Leroy et al., 2024), but remains computationally intensive compared to regression-based methods.

**Geometric Supervision in Pose Learning.** The importance of geometric constraints in pose estimation has been increasingly recognized. (Kendall & Cipolla, 2017) identified the fundamental challenge of balancing rotation and translation components in pose regression. More recently, (Brachmann et al., 2017) and (Brahmbhatt et al., 2018) have shown that incorporating geometric constraints can significantly improve pose estimation accuracy. Additionally, (Wang et al., 2020) showed that camera pose supervision can effectively guide feature descriptor learning through epipolar constraints, enabling training on larger and more diverse datasets. These works establish that ge-

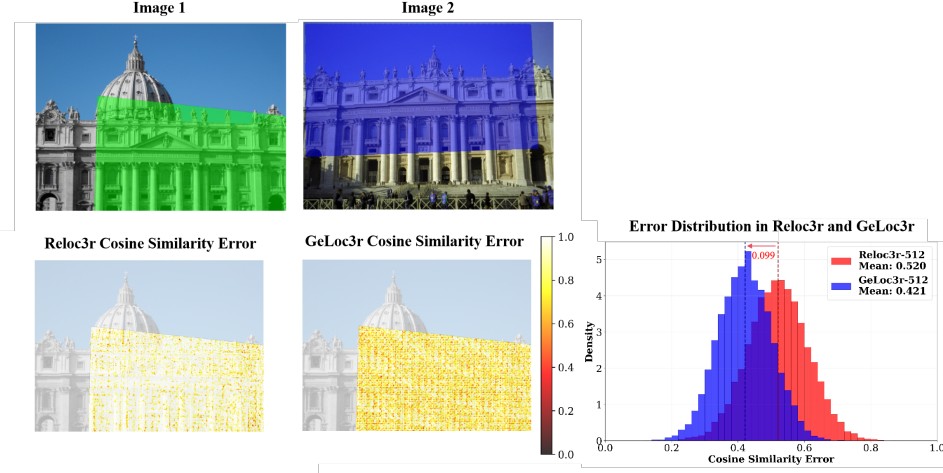

Figure 1: **Cosine Similarity Error Analysis on MegaDepth1500.** Top row: Input image pair with overlapping regions highlighted (green in Image 1, blue in Image 2). Bottom row: Cosine similarity error maps, obtained by projecting MASt3R's pixel-wise descriptor features from Image 2 onto Image 1 using predicted poses of ReLoc3R and GeLoc3r. ReLoc3R (left) exhibits widespread high errors (yellow regions), while GeLoc3r (middle) achieves significantly lower errors through geometric consistency regularization. The last plot shows GeLoc3r reduces mean error from 0.520 to 0.421, shifting the distribution toward lower values and mitigating pose regression inconsistency. Error maps and distributions are normalized for visualization.

ometric constraints provide crucial supervisory signals that pure pose regression lacks, motivating our approach to leverage privileged geometric knowledge during training.

**Hybrid Approaches.** FAR (Rockwell & Fouhey, 2024) demonstrates that combining regression and correspondence-based poses through learned fusion weights improves robustness. FAR employs a Transformer to adaptively balance solved poses (from correspondences) and learned poses (regression) at inference time, prioritizing reliable matches through correspondence weighting and pose interpolation. However, this dual-method inference negates the speed advantage of regression. In contrast, we introduce a fundamentally different paradigm: geometric supervision exists only during training. Our FusionTransformer learns correspondence weights to guide RANSAC-based geometric constraints during training, but disappears entirely at inference. This transfers geometric knowledge into network weights while preserving regression's computational efficiency—achieving geometric accuracy without geometric computation at inference time.

## 3 METHOD

### 3.1 MOTIVATION: ADDRESSING DESCRIPTOR INCONSISTENCY IN POSE REGRESSION

In ReLoc3R, when projecting pixels from Image 1 to Image 2 using its predicted relative pose transformation, we discovered significant cosine similarity errors between MASt3R pretrained descriptor features extracted from corresponding regions. As shown in Figure 1, ReLoc3R's error map is dominated by yellow regions (high error) with very few red points (low error), indicating that pixel-level correspondences are largely inaccurate despite reasonable overall pose estimates. The overlapping regions exhibit widespread descriptor mismatches—geometrically corresponding points that should have similar descriptors instead showing high cosine similarity errors. This inconsistency indicates that regression networks optimize for pose accuracy without ensuring geometric consistency in their learned representations.

This observation directly motivates our first technical innovation: introducing pixel-wise descriptor loss constraints between corresponding image regions during training. By enforcing that projected pixels maintain high cosine similarity between their MASt3R descriptor features (256-dimensional vectors), we teach the network that accurate poses must preserve geometric correspondence relation-

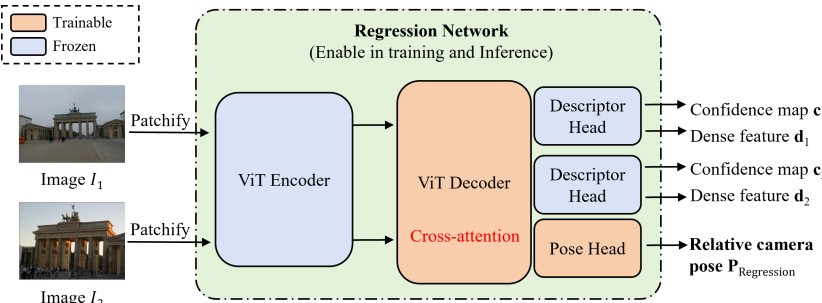

Figure 2: **GeLoc3r Architecture at Inference.** The model processes image pairs $(I_1, I_2)$ through a shared ViT encoder followed by a ViT decoder with cross-attention. Three task-specific heads produce outputs: (1) a trainable pose head generates the relative camera pose $\mathbf{P}_{regression}$, (2-3) two frozen descriptor heads (pre-trained from MASt3R) output dense features $\mathbf{d}_1, \mathbf{d}_2$ and confidence maps $\mathbf{c}_1, \mathbf{c}_2$. At inference, only the pose regression output is used, while the descriptor outputs are only used during training.

ships. This constraint ensures that the predicted transformation aligns not just in pose space but also in the learned feature space. To further improve robustness, we develop a weighted RANSAC approach that learns to identify and prioritize reliable correspondences through a FusionTransformer module. This advanced solution filters out unreliable matches caused by occlusions or texture-less regions, focusing the geometric supervision on high-confidence correspondences that provide stronger learning signals.

Figure 1 validates our approach, with GeLoc3r reducing mean error from 0.520 to 0.421, demonstrating that GCR successfully addresses the pixel-level matching inaccuracy in pose regression.

It reveals an important problem: **regression networks trained with only pose loss never learn that camera poses must be geometrically consistent with 3D structure**. It minimizes $\|\mathbf{P}_{pred} - \mathbf{P}_{gt}\|$ without ensuring that descriptors maintain geometric correspondence across views. This explains why regression methods consistently underperform correspondence-based approaches that explicitly enforce geometric constraints. Our solution revolutionizes the training paradigm: instead of relying solely on pose supervision, we introduce Geometric Consistency Regularization (GCR)—a training framework that teaches regression networks to understand geometry through multi-level supervision. The key innovation is that GCR exists **only during training**, using privileged information (GT depth) to generate geometric constraints that transfer knowledge into the network. At inference, the model runs as regression only method, maintaining fast speed while benefiting from the geometric understanding learned during training.

### 3.2 ARCHITECTURE OVERVIEW: A NOVEL TRAINING FRAMEWORK

At inference (Figure 2), GeLoc3r operates identically to ReLoc3R—a regression method for fast speed. During training (Figure 3), we revolutionize the learning process by introducing GCR, which extracts dense descriptors using frozen pre-trained MASt3R heads, generates 3D-2D correspondences from ground-truth depth, weights them via a learned FusionTransformer, and computes geometrically-consistent poses through weighted RANSAC to enforce consistency between regression and geometric predictions. The key is that **all geometric components (FusionTransformer, RANSAC, GT depth) exist only during training**, which teaches the regression network to produce geometrically-consistent poses. At inference, only the enhanced regression head is used, maintaining fast speed while benefiting from the geometric knowledge learned during training.

### 3.3 REGRESSION NETWORK BACKBONE WITH ENCODER-DECODER

We employ a Siamese encoder with shared weights to extract features from both images. Each image $I_i$ is divided into 16×16 patches and projected to dimension 1024 with 2D Rotary Positional Embeddings (RoPE) (Su et al., 2021). The decoder uses bidirectional cross-attention to establish

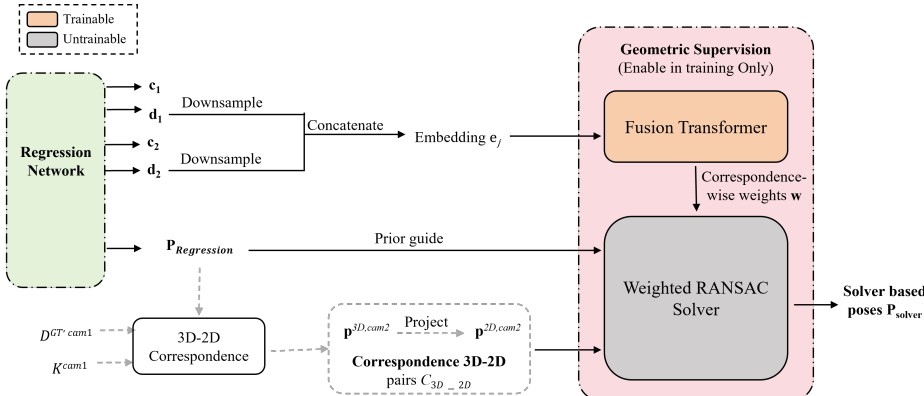

Figure 3: **GeLoc3r Training with Geometric Consistency Regularization (GCR).** During training, the model leverages ground-truth depth maps and camera intrinsics as privileged information. Dense features from frozen descriptor heads are downsampled and concatenated to form correspondence embeddings $\mathbf{e}_j$, which are processed by the FusionTransformer to produce per-correspondence weights $\mathbf{w}$. Simultaneously, GT depth is unprojected to 3D points, transformed using the predicted pose $\mathbf{P}_{regression}$, and projected to form 3D-2D correspondence pairs $\mathcal{C}_{3D-2D}$ (detailed formation process in Appendix A.2). The weighted RANSAC solver uses these correspondences with FusionTransformer weights and the regression pose as a prior to compute $\mathbf{P}_{solver}$. The consistency loss between $\mathbf{P}_{regression}$ and $\mathbf{P}_{solver}$ provides geometric supervision that teaches the regression network to produce geometrically consistent poses. The pink background highlights the training-only GCR module that is not used during inference.

implicit correspondences:

$$\mathbf{f}_i^{enc} = \text{Encoder}(I_i + \text{RoPE}(p_i)), \quad i \in \{1, 2\} \tag{1}$$

$$\mathbf{f}_1^{dec}, \mathbf{f}_2^{dec} = \text{CrossDecoder}(\mathbf{f}_1^{enc}, \mathbf{f}_2^{enc}) \tag{2}$$

This cross-attention mechanism enables features from one image to query relevant features in the other, creating representations that capture both appearance and geometric relationships crucial for pose estimation. At inference, these features are directly passed to the pose head for prediction (Figure 2), while during training they are additionally feed into the GCR module for geometric supervision.

### 3.4 POSE HEAD AND DESCRIPTOR HEAD

The prediction heads operate on concatenated encoder and decoder features to leverage both local and global information.

**Pose Head.** A shared pose head processes decoder features to predict relative transformations. For rotation, it predicts a 9D representation (flattened 3×3 matrix) converted to SO(3) via SVD (Zhou et al., 2019). For translation, it predicts a normalized direction $\hat{\mathbf{t}}$ and scale $s \in [0, s_{max}]$:

$$\mathbf{P}_{\text{regression}} = \text{PoseHead}(\mathbf{f}^{dec}), \quad \mathbf{t} = s \cdot \hat{\mathbf{t}} \tag{3}$$

**Descriptor Head.** Frozen descriptor heads (pre-trained from MASt3R) extract 24-dimensional descriptors and confidence maps:

$$\mathbf{d}_i, \mathbf{c}_i = \text{DescHead}([\mathbf{f}_i^{enc}; \mathbf{f}_i^{dec}]), \quad i \in \{1, 2\} \tag{4}$$

These provide geometric anchors for training-time supervision.

### 3.5 CORRESPONDENCE-WISE WEIGHT

Our FusionTransformer is a specialized attention module that learns to assess correspondence quality for geometric supervision. It consists of two standard transformer blocks with hidden dimension

768 and 4 attention heads, processing concatenated descriptor pairs through self-attention layers to produce per-correspondence weights that guide RANSAC sampling. The key innovation is using attention mechanisms to model inter-correspondence relationships—correspondences are not evaluated independently but in context of all other correspondences, enabling the network to identify consistent geometric patterns and suppress outliers.

The FusionTransformer operates solely on descriptor correspondences without directly using pose predictions as input. Instead, it learns to assess correspondence quality through self-attention, exploiting the geometric principle that valid correspondences exhibit consistent transformation patterns while outliers deviate from this consensus. It first computes putative correspondences by subsampling descriptors with appropriate stride $s$ and concatenate them to form correspondence embeddings:

$$\mathbf{e}_j = [\mathbf{d}_{1,j}; \mathbf{d}_{2,j}] \in \mathbb{R}^{48} \tag{5}$$

The FusionTransformer processes these through self-attention to produce normalized weights:

$$\mathbf{w} = \mathrm{Softmax}(\mathrm{MLP}(\mathrm{FusionTransformer}(\mathbf{e}))) \tag{6}$$

where $\sum_i w_i = 1$. These weights guide RANSAC sampling during training to prioritize geometrically consistent correspondences. Unlike traditional matching that requires explicit feature matching, it leverages the cross-attention decoder's implicit spatial alignment: descriptors at the same spatial location in both views (after subsampling) are treated as potential correspondences. Specifically, for each spatial position $i$ in the subsampled grid, it pairs $\mathbf{d}_1^{\mathrm{sub}}[i]$ with $\mathbf{d}_2^{\mathrm{sub}}[i]$. This approach provides dense correspondences while maintaining computational tractability.

### 3.6 GEOMETRIC CONSISTENCY REGULARIZATION (GCR) DURING TRAINING

GCR regularizes regression network learning by introducing geometric supervision that was previously not explored. It operates through a chain of implications: **correct pose → accurate projections → matching descriptors**. We leverage this chain to create multiple regularizations for training dynamics.

**The Consistency Chain:** Our framework follows a logical progression where if $\mathbf{P}_{\mathrm{regression}}$ is geometrically correct, it should transform 3D points accurately between views. If transformations are accurate, projected points should land at correct pixel locations. If projections are correct, descriptors at corresponding locations should match. This motivates our dual supervision strategy: Direct and Indirect Supervision.

**Direct Supervision - Pose Consistency:** Our framework directly verifies pose quality by comparing with a geometrically-solved pose. To generate $\mathbf{P}_{solver}$, it employs a weighted RANSAC algorithm that leverages both GT depth-derived 3D-2D correspondences and FusionTransformer weights. We extend standard RANSAC (Fischler & Bolles, 1981) with two key enhancements: learned importance sampling and prior-guided hypothesis scoring. The weighted RANSAC operates as follows: (1) samples 6-point minimal sets with probability $p_i = w_i / \sum_j w_j$, where weights $w_i$ are learned by the FusionTransformer from descriptor correspondences, biasing selection toward reliable matches; (2) solves the PnP problem (Lepetit et al., 2009) to obtain candidate pose $\mathbf{P}_{candidate}$; and (3) evaluates each hypothesis using a hybrid scoring function Score $= N_{inliers} + \beta \cdot \exp(-||\mathbf{P}_{candidate} - \mathbf{P}_{\mathrm{regression}}||_F / \tau)$. This scoring combines classical geometric verification (inlier counting with 2-pixel threshold) with a soft prior that favors poses near the regression prediction, using $\beta = 0.5$ to balance both terms and $\tau = 10$ for the exponential decay rate. After 50 iterations, the highest-scoring pose becomes $\mathbf{P}_{solver}$. The complete weighted RANSAC algorithm is detailed in Appendix A.1.

The consistency loss then measures the angular error between the regression and solver poses:

$$\mathcal{L}_{consistency} = \mathrm{AngularError}(\mathbf{P}_{\mathrm{regression}}, \mathbf{P}_{solver}) \tag{7}$$

This teaches the network that good poses must satisfy geometric constraints. Importantly, better $\mathbf{P}_{\mathrm{regression}}$ predictions lead to more consistent correspondences for RANSAC supervision, creating a positive feedback loop during training.

**Indirect Supervision - Projection Verification:** The framework tests pose quality through its projections. A correct pose should project 3D points to locations with matching descriptors:

$$\mathcal{L}_{descriptor} = -\sum_i w_i \cdot \mathrm{sim}(\mathbf{d}_i^{src}, \mathbf{d}_i^{proj}) \tag{8}$$

where $\mathbf{d}_i^{proj} = \text{Descriptor}(\Pi(\mathbf{P}_{\text{regression}} \cdot \mathbf{p}_i^{3D}))$ and $w_i$ are confidence-based weights derived from the descriptor confidence maps $\mathbf{c}_i$ to focus on reliable regions. The benefit of this approach is **descriptor similarity serves as a verification mechanism for pose quality**. If the regression pose is wrong, it will project points to incorrect locations where descriptors don't match, creating a strong error signal. This teaches the network that correct poses must not only minimize pose error but also maintain correspondence consistency across views.

The final training objective is

$$\mathcal{L}_{total} = \lambda_{pose}\mathcal{L}_{pose} + \lambda_{consistency}\mathcal{L}_{consistency} + \lambda_{desc}\mathcal{L}_{descriptor} \tag{9}$$

Note that while $\mathcal{L}_{pose}$ exists in all regression methods, the latter two terms are our proposed GCR that exists only during training. This creates a self-reinforcing learning framework where the network learns that geometrically correct poses lead to accurate projections, which in turn produce high descriptor similarities. The descriptor loss verifies whether the predicted pose correctly establishes correspondences between views—a fundamental requirement for valid camera pose estimation. This training paradigm transfers geometric knowledge into the network weights, enabling the model to produce geometrically consistent poses at inference without any additional computation.

## 4 EXPERIMENTS

We conduct extensive experiments to evaluate GeLoc3r's performance against SOTA visual localization methods. Our evaluations focus on demonstrating that geometric consistency regularization during training leads to improved pose accuracy at inference while maintaining the efficiency of regression-based approaches.

### 4.1 SETUP

**Training Data.**

GeLoc3r is trained on 10 diverse datasets including CO3Dv2 (Reizenstein et al., 2021), ScanNet++ (Yeshwanth et al., 2023), ARKitScenes (Baruch et al., 2021), MegaDepth (Li & Snavely, 2018), Waymo Open (Sun et al., 2020), StaticThings3D (Leroy et al., 2023), Habitat (Savva et al., 2019), BlendedMVS (Yao et al., 2020), RealEstate10K, and WildRGB-D (Leroy et al., 2023), sampling 50,000 image pairs from each per epoch. All datasets provide ground-truth depth essential for GCR supervision. We resize images to resolutions between (512, 160) and (512, 384) with padding and apply color jittering.

**Implementation Details.** GeLoc3r uses $m = 24$ encoder blocks, $n = 12$ decoder blocks, followed by the pose regression head with $h = 2$ convolutional layers. We initialize GeLoc3r with pre-trained MASt3R's 512-DPT weights (Leroy et al., 2024). The encoder remains frozen during training to preserve the learned visual representations, while the descriptor heads from MASt3R also remain frozen, outputting 24-dimensional descriptors at 1/16 resolution. The trainable components include the decoder, pose head, and FusionTransformer, while the weighted RANSAC solver operates as a non-trainable geometric module. The FusionTransformer uses 2 transformer blocks with hidden dimension 768 and 4 attention heads. During training, we sample 3D-2D correspondences with stride 8 from the dense depth maps. The weighted RANSAC solver runs for 50 iterations using 6-point minimal sets with a 2-pixel reprojection threshold. The consistency loss is applied only when more than 50 valid correspondences are available after filtering.

The total loss combines multiple objectives with weights determined empirically. $\lambda_{pose} = 0.8$ for pose regression, $\lambda_{desc} = 0.1$ for descriptor alignment, and $\lambda_{consistency} = 0.1$ for pose consistency. The full model is trained on 8 NVIDIA L40S GPUs with a batch size of 10 and a learning rate starting at $10^{-5}$, decaying to $10^{-7}$ using cosine annealing. We use AdamW optimizer (Loshchilov & Hutter, 2019) with gradient clipping at 1.0. All evaluations and inference speed measurements are conducted on a single 24GB NVIDIA GeForce RTX 4090 GPU for fair comparison.

### 4.2 RELATIVE CAMERA POSE ESTIMATION

We evaluate GeLoc3r's relative pose estimation performance on standard benchmarks, including ScanNet1500 (Dai et al., 2017; Sarlin et al., 2020), RealEstate10K, MegaDepth1500 (Li & Snavely,

Table 1: Relative camera pose evaluation results on the four benchmarking datasets. We report pose accuracy measured at thresholds of 5/10/20 degrees. The best results for PR method category are highlighted in bold. Results for baseline methods except CO3Dv2 are from (Dong, 2024).

| Methods | ScanNet1500 | | | RealEstate10K | | | MegaDepth1500 | | | CO3Dv2 | | | Time |
| | AUC@5° | AUC@10° | AUC@20° | AUC@5° | AUC@10° | AUC@20° | AUC@5° | AUC@10° | AUC@20° | AUC@5° | AUC@10° | AUC@20° | (ms) |
|---|---|---|---|---|---|---|---|---|---|---|---|---|---|
| *Non-PR Methods* | | | | | | | | | | | | | |
| Efficient LoFTR | 19.20 | 37.00 | 53.60 | - | - | - | 56.4 | 72.2 | 83.5 | - | - | - | 40 |
| ROMA | 28.90 | 50.40 | 68.30 | 54.60 | 69.80 | 79.70 | 62.6 | 76.7 | 86.3 | - | - | - | 300 |
| DUSt3R (Leroy et al., 2023) | 23.81 | 45.91 | 65.57 | 39.70 | 56.88 | 70.43 | 27.9 | 46.0 | 63.3 | - | - | - | 441 |
| MASt3R (Leroy et al., 2024) | 28.01 | 50.24 | 68.83 | 63.54 | 76.39 | 84.50 | 42.4 | 61.5 | 76.9 | - | - | - | 294 |
| NoPoSplat | 31.80 | 53.80 | 71.70 | 69.10 | 80.60 | 87.70 | - | - | - | - | - | - | >2000 |
| *PR Methods* | | | | | | | | | | | | | |
| Map-free (Regress-SN) | 1.84 | 8.75 | 25.33 | 0.83 | 4.06 | 13.97 | - | - | <10 | - | - | - | 10 |
| Map-free (Regress-MF) | 0.50 | 3.48 | 13.15 | 1.61 | 6.74 | 18.38 | - | - | <10 | - | - | - | 10 |
| ExReNet (SN) | 2.30 | 10.71 | 26.13 | 2.17 | 7.94 | 20.43 | - | - | <10 | - | - | - | 17 |
| ExReNet (SUNCG) | 1.61 | 7.00 | 18.03 | 3.27 | 12.06 | 27.85 | - | - | <10 | - | - | - | 17 |
| ReLoc3R-224 (Dong, 2024) | 28.34 | 52.60 | 71.56 | 59.70 | 75.05 | 84.71 | 39.9 | 59.7 | 75.4 | - | - | - | 15 |
| ReLoc3R-512 (Dong, 2024) | **34.79** | **58.37** | **75.56** | 66.70 | 80.20 | 88.39 | 49.6 | 67.9 | 81.2 | 34.85 | 54.39 | 69.90 | 25 |
| **GeLoc3r-512 (Ours)** | 34.40 | 58.12 | 75.56 | **68.66** | **81.49** | **89.18** | **50.45** | **69.29** | **82.55** | **40.45** | **59.80** | **74.31** | 33 |

Table 2: Visual localization results on the 7-Scenes dataset (Shotton et al., 2013), which is entirely unseen during training. We report median pose errors in meters and degrees. Best results are highlighted in bold. Results for baseline methods are from (Dong, 2024).

| Methods | Chess | Fire | Heads | Office | Pumpkin | RedKitchen | Stairs | Average |
|---|---|---|---|---|---|---|---|---|
| EssNet (CL) | - | - | - | - | - | - | - | 0.57/80.06 |
| NC-EssNet (CL) | - | - | - | - | - | - | - | 0.48/32.97 |
| Relative PN (U) | 0.31/15.05 | 0.40/19.00 | 0.24/22.15 | 0.38/14.14 | 0.44/18.24 | 0.41/16.51 | 0.35/23.55 | 0.36/18.38 |
| RelocNet (SN) | 0.21/10.9 | 0.32/11.8 | 0.15/13.4 | 0.31/10.3 | 0.40/10.9 | 0.33/10.3 | 0.33/11.4 | 0.29/11.3 |
| ImageNet+NCM | - | - | - | - | - | - | - | 0.19/4.30 |
| Map-free (Match) | 0.10/2.93 | 0.12/4.95 | 0.11/5.40 | 0.12/3.01 | 0.16/3.19 | 0.14/3.45 | 0.21/4.50 | 0.14/3.92 |
| Map-free (Regress) | 0.09/2.66 | 0.13/4.54 | 0.11/4.81 | 0.11/2.77 | 0.16/3.11 | 0.14/3.48 | 0.18/4.70 | 0.13/3.72 |
| ExReNet (SN) | 0.06/2.15 | 0.09/3.20 | 0.04/3.30 | 0.07/2.17 | 0.11/2.65 | 0.09/2.57 | 0.33/7.34 | 0.11/3.34 |
| ExReNet (SUNCG) | 0.05/1.63 | 0.07/2.54 | 0.03/2.71 | 0.06/1.75 | 0.07/2.04 | 0.07/2.10 | 0.19/4.87 | 0.08/2.52 |
| ReLoc3R-224 | **0.03**/0.99 | 0.04/1.13 | 0.02/1.23 | 0.05/0.88 | 0.07/1.14 | 0.05/1.23 | 0.12/2.25 | 0.05/1.26 |
| ReLoc3R-512 | **0.03**/0.88 | 0.03/**0.81** | **0.01**/0.95 | **0.04**/0.88 | 0.06/1.10 | **0.04**/1.26 | **0.07**/**1.26** | **0.04**/**1.02** |
| **GeLoc3r-512 (Ours)** | **0.03**/**0.80** | **0.02**/0.82 | **0.01**/0.83 | **0.04**/0.82 | 0.05/**1.00** | **0.04**/1.22 | 0.08/2.14 | **0.04**/1.09 |

2018), and CO3Dv2 (Reizenstein et al., 2021), and compare it with state-of-the-art methods across different categories. We adopt the AUC (Area Under Curve) metric at angular thresholds of 5°, 10°, and 20°, which measures the percentage of predictions where the maximum angular error (between rotation and translation errors) falls below the threshold. Table 1 presents comprehensive evaluation results on the datasets. ReLoc3R (Dong, 2024) achieves state-of-the-art performance among pose regression methods, significantly outperforming earlier approaches such as Map-free and ExReNet by large margins. On the challenging MegaDepth1500 dataset with extreme viewpoint changes, ReLoc3R-512 reaches 81.2% AUC@20°, whereas previous regression methods achieve less than 10%, demonstrating its robustness on outdoor landmarks. Our GeLoc3r consistently improves upon ReLoc3R across all datasets, including the CO3Dv2, where we achieve 40.45% AUC@5° compared to ReLoc3R's 34.85%, validating that geometric consistency regularization enhances performance across diverse scene types. Additional evaluation on BlendedMVS (see Appendix B) further confirms these consistent improvements. Although non-PR methods such as ROMA achieve higher accuracy on certain datasets, they require substantially more computation time (300–2000 ms), making them impractical for real-time applications. In contrast, our method runs in 33 ms, which is over 60× faster.

## 4.3 Visual Localization

In this section, we evaluate GeLoc3r's pose estimation capability on visual localization task. We conduct experiments on two public datasets: 7-Scenes (Shotton et al., 2013) for indoor localization and Cambridge Landmarks (Kendall et al., 2015) for outdoor scenarios. Following the literature, we apply NetVLAD for image retrieval and use the top 10 similar image pairs. We directly use these retrieved image pairs without distance-based clustering or other heuristics. For evaluation, we report the median translation error (in meters) and median rotation error (in degrees) for each scene, following standard protocols that measure the absolute pose accuracy of the localized camera. It is worth noting that both datasets were entirely unseen by GeLoc3r during training, placing our method in the most challenging *unseen* Relative Pose Regression (RPR) task. For more evaluation results in Absolute Pose Regression task and RPR (seen), please check Appendix C.

Table 3: Visual localization results on Cambridge Landmarks (Kendall et al., 2015) for unseen methods. We report median pose errors in meters and degrees. Best results are highlighted in bold. Results for baseline methods are from (Dong, 2024).

| Methods | GreatCourt | KingsCollege | OldHospital | ShopFacade | StMarysChurch | Average (4) | Average |
|---|---|---|---|---|---|---|---|
| EssNet (7S) | - | - | - | - | - | 10.36/85.75 | - |
| NC-EssNet (7S) | - | - | - | - | - | 7.98/24.35 | - |
| Map-free (Match) | 9.09/5.33 | 2.51/3.11 | 3.89/6.44 | 1.04/3.61 | 3.00/6.14 | 2.61/4.83 | 3.90/4.93 |
| Map-free (Regress) | 8.40/4.56 | 2.44/2.54 | 3.73/5.23 | 0.97/3.17 | 2.91/5.10 | 2.51/4.01 | 3.69/4.12 |
| ExReNet (SN) | 10.97/6.52 | 2.48/2.92 | 3.47/3.90 | 0.90/3.27 | 2.60/4.98 | 2.36/3.77 | 4.08/4.32 |
| ExReNet (SUNCG) | 9.79/4.46 | 2.33/2.48 | 3.54/3.49 | 0.72/2.41 | 2.30/3.72 | 2.22/3.03 | 3.74/3.31 |
| ImageNet+NCM | - | - | - | - | - | 0.83/1.36 | - |
| ReLoc3R-224 | 1.71/0.94 | 0.47/0.41 | 0.87/0.66 | 0.18/**0.53** | 0.41/0.73 | 0.48/0.58 | 0.73/0.65 |
| ReLoc3R-512 | 1.22/0.73 | 0.42/0.36 | **0.62**/0.55 | **0.13**/0.58 | 0.34/**0.58** | 0.38/**0.52** | 0.55/0.56 |
| **GeLoc3r-512 (Ours)** | **1.07/0.52** | **0.41/0.34** | 0.64/**0.54** | 0.15/0.64 | **0.27/0.58** | **0.37**/0.53 | **0.51/0.52** |

**Indoor visual localization.** Table 2 presents results for unseen RPR methods on 7-Scenes, where all methods are evaluated without any dataset-specific training. This represents the most challenging evaluation scenario, testing true generalization capability. Methods like EssNet and NC-EssNet show catastrophic failure with errors exceeding 30°, while more recent approaches achieve progressively better results. ReLoc3R (Dong, 2024) demonstrates strong performance with 0.04m/1.02° average error, and our GeLoc3r achieves comparable results at 0.04m/1.09°, maintaining the same translation accuracy across all unseen methods.

**Outdoor visual localization.** Table 3 shows results for unseen RPR methods on Cambridge Landmarks, where outdoor environments pose additional challenges. Early methods like EssNet and NC-EssNet again fail dramatically, while Map-free and ExReNet methods achieve errors around 3-4 meters. ReLoc3R (Dong, 2024) achieves breakthrough performance with 0.55m/0.56° average error, representing over 6× improvement compared to previous best unseen methods. Our GeLoc3r further improves to 0.51m/0.52°, demonstrating that geometric consistency regularization enhances localization accuracy even in challenging outdoor scenarios.

To further demonstrate the benefits of GeLoc3r on extended trajectories, we visualize the localization results on the Cambridge Landmarks GreatCourt test sequence containing 612 poses spanning a 38.52m trajectory. Figure 4 provides a direct visual comparison between ReLoc3r and GeLoc3r. As shown in Figure 4(a), vanilla ReLoc3r exhibits numerous large-error outliers (indicated by long red error lines), with errors ranging from 0.040m to 123.508m (mean: 5.096m). In contrast, Figure 4(b) demonstrates that GeLoc3r produces visibly tighter alignment to ground truth with significantly fewer large-error outliers, achieving errors ranging from 0.053m to 52.379m (mean: 2.826m). Quantitatively, after Sim3 alignment (rotation, translation, and scale), GeLoc3r achieves an RMSE of 6.411m compared to ReLoc3r's 10.862m, representing a 41% reduction. Additionally, we observe a 58% reduction in maximum error and a 45% reduction in mean error. The error lines connecting predicted positions to ground truth clearly illustrate that GeLoc3r maintains consistent accuracy across the entire trajectory, validating that our geometric consistency regularization effectively reduces catastrophic localization failures in extended sequences.

## 4.4 ABLATION STUDIES

We conduct ablation studies on ScanNet to analyze the contribution of each component in our GCR framework. Table 4 shows the progressive improvements from our baseline to the full model. All models are initialized from MASt3R's pre-trained 512-DPT checkpoint.

Table 4: Ablation study on ScanNet. Each component of GCR contributes to improved pose accuracy. All models start from MASt3R 512-DPT pre-trained weights.

| Method | AUC@5° | AUC@10° | AUC@20° |
|---|---|---|---|
| Baseline (pose regression only) | 19.4 | 44.9 | 66.6 |
| + Descriptor loss | 20.3 | 46.3 | 67.5 |
| + Weighted RANSAC consistency (Full GeLoc3r) | **22.4** | **48.3** | **68.8** |

The results demonstrate that each component of our Geometric Consistency Regularization (GCR) contributes meaningfully to performance. Starting from the MASt3R-initialized baseline with pose

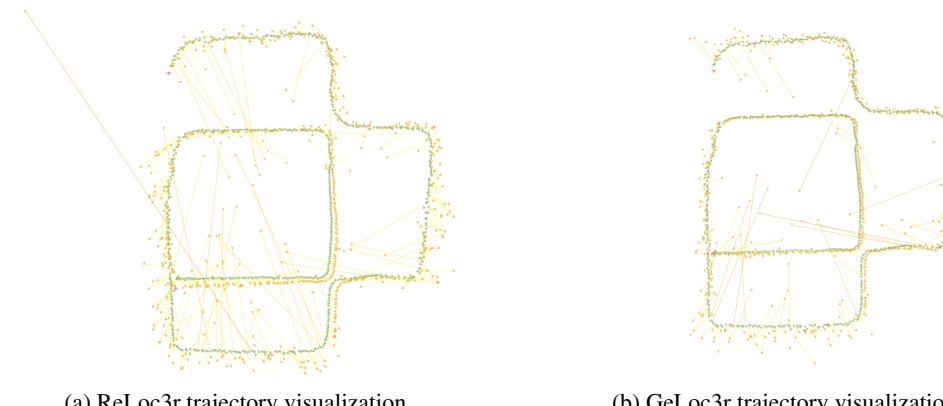

(a) ReLoc3r trajectory visualization        (b) GeLoc3r trajectory visualization

Figure 4: **Visual localization trajectory comparison on Cambridge Landmarks GreatCourt.** We visualize localization results on 612 test poses spanning a 38.52m trajectory. The green line shows the ground truth camera path. Orange points represent predicted camera positions. **Error lines** connect each predicted pose to its corresponding ground truth position, where **line length visualizes the error magnitude**. (a) ReLoc3r exhibits numerous large-error outliers with errors ranging from 0.040m to **123.508m** (mean: 5.096m). (b) GeLoc3r achieves visibly tighter alignment with errors ranging from 0.053m to **52.379m** (mean: 2.826m), representing a **58% reduction in maximum error** and a **45% reduction in mean error**. The visualization clearly demonstrates that our geometric consistency regularization effectively mitigates extreme outliers in extended sequences.

regression only (19.4% AUC@5°), adding the confidence-weighted descriptor loss improves to 20.3% AUC@5°, while incorporating weighted RANSAC pose consistency further improves to 22.4% AUC@5°. The full GeLoc3r model achieves a total improvement of 3.0 percentage points in AUC@5° over the baseline, validating our hypothesis that geometric supervision during training enhances the accuracy of pose regression networks.

## 5 CONCLUSION

We introduce GeLoc3r, a novel framework for relative pose estimation that addresses fundamental key limitations of regression-based approaches through Geometric Consistency Regularization (GCR). By leveraging geometric supervision during training, GeLoc3r improves regression accuracy without compromising inference efficiency. Our core idea is to apply RANSAC-based geometric constraints when ground-truth depth is available, while relying on learned geometric priors for robust regression at inference. The FusionTransformer learns correspondence quality and confidence, enabling effective geometric supervision via consistency regularization. GCR achieves the accuracy of geometric methods with the speed of pose regression, showing that privileged geometric knowledge can substantially enhance regression networks. Experiments demonstrate state-of-the-art performance with superior robustness and fast inference. Future work will extend GCR to multi-view settings and explore alternative geometric supervision strategies for visual localization.

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

# A  IMPLEMENTATION DETAILS

## A.1  WEIGHTED RANSAC ALGORITHM - FULL DETAILS

The weighted RANSAC algorithm used to generate $\mathbf{P}_{solver}$ (described in Section 3.5) is formalized below:

---
**Algorithm 1** Weighted RANSAC with Prior Guidance

---
**Require:** 3D-2D correspondences $\mathcal{C}_{3D-2D}$, weights $\mathbf{w}$, prior $\mathbf{P}_{\text{regression}}$
**Ensure:** Refined pose $\mathbf{P}_{solver}$
 1: **for** $iter = 1$ to 50 **do**
 2:    Sample 6 correspondences using probability $p_i = w_i / \sum_j w_j$
 3:    Solve PnP to get $\mathbf{P}_{candidate}$
 4:    Compute inliers with reprojection error $< 2$ pixels
 5:    Score $= N_{inliers} + \beta \cdot \exp(-||\mathbf{P}_{candidate} - \mathbf{P}_{\text{regression}}||_F / \tau)$
 6:    Update best if score improves
 7: **end for**
 8: **return** Best scoring pose

---

The algorithm enhances standard RANSAC through: (1) importance-weighted sampling based on FusionTransformer weights, and (2) prior-guided scoring that balances geometric evidence with the regression pose prior. As discussed in the main text, $\beta = 0.5$ and $\tau = 10$ were determined through ablation studies.

### A.2 3D-2D CORRESPONDENCE FORMATION

The correspondence pipeline transforms GT depth to 3D-2D pairs:

$$P^{3D} = \text{Unproject}(D^{\text{GT,cam1}}, K^{\text{cam1}}) \qquad (10)$$

$$\mathbf{p}_i^{3D,\text{cam2}} = \mathbf{P}_{\text{Regression}} \cdot [\mathbf{p}_i^{3D,\text{cam1}}; 1]^T \qquad (11)$$

$$\mathbf{p}_i^{2D,\text{cam2}} = \text{Project}(\mathbf{p}_i^{3D,\text{cam2}}) \qquad (12)$$

We sample points on a regular grid with stride $s = 8$. The quality depends on $\mathbf{P}_{\text{Regression}}$ accuracy—better pose predictions yield more consistent correspondences for RANSAC supervision.

## B ADDITIONAL RELATIVE POSE ESTIMATION RESULTS

### B.1 BLENDEDMVS EVALUATION

We provide additional evaluation on the BlendedMVS dataset (Yao et al., 2020), which contains multi-view images reconstructed from internet photos covering diverse scenes including sculptures, buildings, and statues.

Table 5: Relative pose estimation results on BlendedMVS dataset. We report pose accuracy at thresholds of 5/10/20 degrees. Best results are highlighted in bold.

| Method | AUC@5° | AUC@10° | AUC@20° |
|---|---|---|---|
| ReLoc3R-512 | 69.09 | 81.69 | 89.61 |
| **GeLoc3r-512 (Ours)** | **72.17** | **83.51** | **90.67** |

As shown in Table 5, GeLoc3r consistently outperforms ReLoc3R on BlendedMVS, achieving improvements of 3.08% in AUC@5°, demonstrating that our Geometric Consistency Regularization effectively enhances pose estimation accuracy on this challenging multi-view reconstruction dataset.

## C COMPLETE VISUAL LOCALIZATION RESULTS

This section provides complete evaluation results including Absolute Pose Regression (APR) and Relative Pose Regression (RPR) methods with both seen and unseen settings.

Table 6: Complete visual localization results on 7-Scenes dataset (Shotton et al., 2013). We report median pose errors in meters and degrees. Methods are categorized into APR, RPR (Seen) trained on 7-Scenes, and RPR (Unseen) evaluated without dataset-specific training. Results for all baseline methods are from (Dong, 2024).

| Methods | Chess | Fire | Heads | Office | Pumpkin | RedKitchen | Stairs | Average |
|---|---|---|---|---|---|---|---|---|
| *APR* | | | | | | | | |
| LENS | 0.03/1.30 | 0.10/3.70 | 0.07/5.80 | 0.07/1.90 | 0.08/2.20 | 0.09/2.20 | 0.14/3.60 | 0.08/3.00 |
| PMNet | 0.03/1.26 | 0.04/1.76 | **0.02**/1.68 | 0.06/1.69 | 0.07/1.96 | 0.08/2.23 | 0.11/2.97 | 0.06/1.93 |
| DFNet+NeFeS | **0.02/0.57** | **0.02/0.74** | 0.02/1.28 | **0.02/0.56** | **0.02/0.55** | **0.02/0.57** | **0.05/1.28** | **0.02/0.79** |
| Marepo | **0.02**/1.24 | **0.02**/1.39 | 0.02/2.03 | 0.03/1.26 | 0.04/1.48 | 0.04/1.71 | 0.06/1.67 | 0.03/1.54 |
| *RPR (Seen)* | | | | | | | | |
| EssNet (7S) | - | - | - | - | - | - | - | 0.22/8.03 |
| Relative PN (7S) | 0.13/6.46 | 0.26/12.72 | 0.14/12.34 | 0.21/7.35 | 0.24/6.35 | 0.24/8.03 | 0.27/11.82 | 0.21/9.30 |
| NC-EssNet (7S) | - | - | - | - | - | - | - | 0.21/7.50 |
| RelocNet (7S) | 0.12/4.14 | 0.26/10.4 | 0.14/10.5 | 0.18/5.32 | 0.26/4.17 | 0.23/5.08 | 0.28/7.53 | 0.21/6.73 |
| Relpose-GNN | 0.08/**2.70** | 0.21/7.50 | 0.13/8.70 | 0.15/4.10 | 0.15/3.50 | 0.19/3.70 | 0.22/6.50 | 0.16/5.20 |
| AnchorNet | 0.06/3.89 | 0.15/10.3 | 0.08/10.9 | 0.09/5.15 | 0.10/2.97 | 0.08/4.68 | 0.10/9.26 | 0.09/6.74 |
| CamNet | **0.04**/1.73 | **0.03/1.74** | **0.05/1.98** | **0.04/1.62** | **0.04/1.64** | **0.04/1.63** | **0.04/1.51** | **0.04/1.69** |
| *RPR (Unseen)* | | | | | | | | |
| EssNet (CL) | - | - | - | - | - | - | - | 0.57/80.06 |
| NC-EssNet (CL) | - | - | - | - | - | - | - | 0.48/32.97 |
| Relative PN (U) | 0.31/15.05 | 0.40/19.00 | 0.24/22.15 | 0.38/14.14 | 0.44/18.24 | 0.41/16.51 | 0.35/23.55 | 0.36/18.38 |
| RelocNet (SN) | 0.21/10.9 | 0.32/11.8 | 0.15/13.4 | 0.31/10.3 | 0.40/10.9 | 0.33/10.3 | 0.33/11.4 | 0.29/11.3 |
| ImageNet+NCM | - | - | - | - | - | - | - | 0.19/4.30 |
| Map-free (Match) | 0.10/2.93 | 0.12/4.95 | 0.11/5.40 | 0.12/3.01 | 0.16/3.19 | 0.14/3.45 | 0.21/4.50 | 0.14/3.92 |
| Map-free (Regress) | 0.09/2.66 | 0.13/4.54 | 0.11/4.81 | 0.11/2.77 | 0.16/3.11 | 0.14/3.48 | 0.18/4.70 | 0.13/3.72 |
| ExReNet (SN) | 0.06/2.15 | 0.09/3.20 | 0.04/3.30 | 0.07/2.17 | 0.11/2.65 | 0.09/2.57 | 0.33/7.34 | 0.11/3.34 |
| ExReNet (SUNCG) | 0.05/1.63 | 0.07/2.54 | 0.03/2.71 | 0.06/1.75 | 0.07/2.04 | 0.07/2.10 | 0.19/4.87 | 0.08/2.52 |
| ReLoc3R-224 | **0.03**/0.99 | 0.04/1.13 | 0.02/1.23 | 0.05/0.88 | 0.07/1.14 | 0.05/1.23 | 0.12/2.25 | 0.05/1.26 |
| ReLoc3R-512 | **0.03**/0.88 | 0.03/**0.81** | **0.01**/0.95 | **0.04**/0.88 | 0.06/1.10 | **0.04**/1.26 | **0.07/1.26** | **0.04/1.02** |
| GeLoc3r-512 (Ours) | **0.03/0.80** | **0.02**/0.82 | **0.01/0.83** | **0.04/0.82** | **0.05/1.00** | **0.04**/1.22 | 0.08/2.14 | **0.04**/1.09 |

Table 7: Complete visual localization results on Cambridge Landmarks (Kendall et al., 2015). We report median pose errors in meters and degrees. Methods are categorized into APR, RPR (Seen) trained on Cambridge, and RPR (Unseen) evaluated without dataset-specific training. Results for all baseline methods are from (Dong, 2024).

| Methods | GreatCourt | KingsCollege | OldHospital | ShopFacade | StMarysChurch | Average (4) | Average |
|---|---|---|---|---|---|---|---|
| *APR* | | | | | | | |
| LENS | - | 0.33/**0.50** | **0.44**/0.90 | 0.27/1.60 | 0.53/1.60 | 0.39/1.20 | - |
| PMNet | - | **0.31**/0.55 | **0.44**/0.79 | 0.17/0.86 | **0.31/0.96** | **0.31**/0.79 | - |
| DFNet+NeFeS | - | 0.37/0.54 | 0.52/0.88 | **0.15/0.53** | 0.37/1.14 | 0.35/**0.77** | - |
| *RPR (Seen)* | | | | | | | |
| EssNet (CL) | - | - | - | - | - | 1.08/3.41 | - |
| Relpose-GNN | 3.20/2.20 | **0.48**/1.00 | **1.14/2.50** | **0.48**/2.50 | 1.52/3.20 | 0.91/2.30 | 1.37/2.30 |
| NC-EssNet (CL) | - | - | - | - | - | 0.85/2.82 | - |
| AnchorNet | - | 0.57/**0.88** | 1.21/2.55 | 0.52/**2.27** | **1.04/2.69** | **0.84/2.10** | - |
| *RPR (Unseen)* | | | | | | | |
| EssNet (7S) | - | - | - | - | - | 10.36/85.75 | - |
| NC-EssNet (7S) | - | - | - | - | - | 7.98/24.35 | - |
| Map-free (Match) | 9.09/5.33 | 2.51/3.11 | 3.89/6.44 | 1.04/3.61 | 3.00/6.14 | 2.61/4.83 | 3.90/4.93 |
| Map-free (Regress) | 8.40/4.56 | 2.44/2.54 | 3.73/5.23 | 0.97/3.17 | 2.91/5.10 | 2.51/4.01 | 3.69/4.12 |
| ExReNet (SN) | 10.97/6.52 | 2.48/2.92 | 3.47/3.90 | 0.90/3.27 | 2.60/4.98 | 2.36/3.77 | 4.08/4.32 |
| ExReNet (SUNCG) | 9.79/4.46 | 2.33/2.48 | 3.54/3.49 | 0.72/2.41 | 2.30/3.72 | 2.22/3.03 | 3.74/3.31 |
| ImageNet+NCM | - | - | - | - | - | 0.83/1.36 | - |
| ReLoc3R-224 | 1.71/0.94 | 0.47/0.41 | 0.87/0.66 | 0.18/**0.53** | 0.41/0.73 | 0.48/0.58 | 0.73/0.65 |
| ReLoc3R-512 | 1.22/0.73 | 0.42/0.36 | **0.62**/0.55 | **0.13**/0.58 | 0.34/**0.58** | 0.38/**0.52** | 0.55/0.56 |
| **GeLoc3r-512 (Ours)** | **1.07/0.52** | **0.41/0.34** | 0.64/**0.54** | 0.15/0.64 | **0.27/0.58** | **0.37**/0.53 | **0.51/0.52** |

# D TRAINING STABILITY AND ERROR BARS

To demonstrate the stability of our training procedure, we trained three independent GeLoc3r models using identical configurations on different machines. We report evaluation results from all runs along with mean and standard deviation (N=3) in Figure 5 and Figure 6.

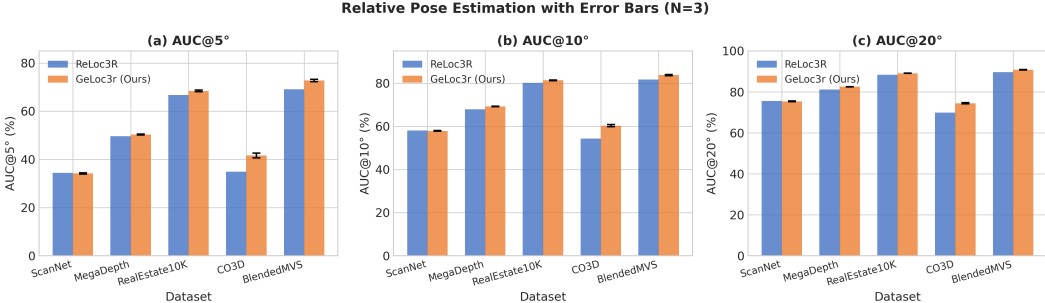

Figure 5: **Relative pose estimation with error bars (N=3 independent training runs).** We compare GeLoc3r against ReLoc3R on AUC@5°, AUC@10°, and AUC@20° across five benchmarks. Error bars indicate ±1 standard deviation. GeLoc3r consistently outperforms ReLoc3R on MegaDepth, RealEstate10K, CO3Dv2, and BlendedMVS with small variance, demonstrating stable and reproducible improvements.

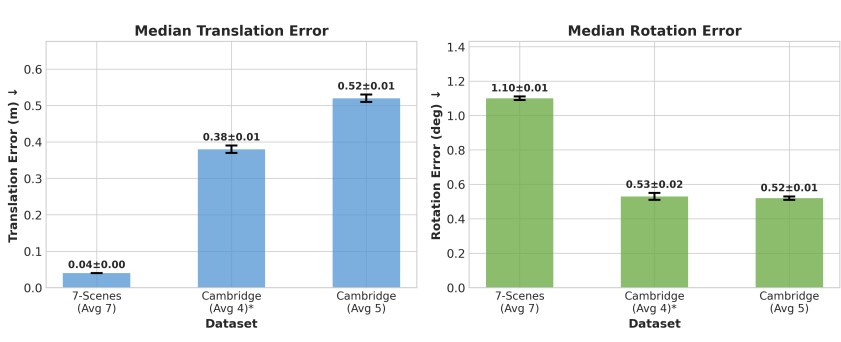

Figure 6: **Visual localization with error bars (N=3 independent training runs).** We report median translation error (m) and rotation error (deg) on 7-Scenes and Cambridge Landmarks. Error bars indicate ±1 standard deviation. The extremely small variance (typically ≤0.02) confirms training stability for visual localization tasks.

The standard deviations are small across all benchmarks (typically <1%), demonstrating stable training behavior. The improvements over ReLoc3R are consistent across all runs. For instance, on CO3D AUC@5, all three runs (40.45, 42.36, 41.97) exceed ReLoc3R (34.85%) with mean improvement of +6.74%. These results confirm that our reported improvements are reproducible and attributable to the proposed method rather than training randomness.

## D.1 DETAILED RESULTS FROM MULTIPLE TRAINING RUNS

Table 8 presents the complete results for relative pose estimation across all three training runs. Table 9 shows the results for absolute pose regression on visual localization benchmarks.

Table 8: Relative pose estimation results from N=3 independent training runs. We report AUC at 5°/10°/20° thresholds (%) ↑.

| Dataset | Metric | Run 1 | Run 2 | Run 3 | Mean $\pm$ Std |
|---|---|---|---|---|---|
| ScanNet | AUC@5 | 34.40 | 34.20 | 33.89 | 34.16 $\pm$ 0.26 |
| | AUC@10 | 58.12 | 57.90 | 57.74 | 57.92 $\pm$ 0.19 |
| | AUC@20 | 75.56 | 75.35 | 75.24 | 75.38 $\pm$ 0.16 |
| MegaDepth | AUC@5 | 50.45 | 50.07 | 50.45 | 50.32 $\pm$ 0.22 |
| | AUC@10 | 69.29 | 69.09 | 69.29 | 69.22 $\pm$ 0.12 |
| | AUC@20 | 82.55 | 82.43 | 82.55 | 82.51 $\pm$ 0.07 |
| RealEstate10K | AUC@5 | 68.66 | 68.54 | 68.04 | 68.41 $\pm$ 0.33 |
| | AUC@10 | 81.49 | 81.44 | 81.15 | 81.36 $\pm$ 0.18 |
| | AUC@20 | 89.18 | 89.17 | 89.02 | 89.12 $\pm$ 0.09 |
| CO3Dv2 | AUC@5 | 40.45 | 42.36 | 41.97 | 41.59 $\pm$ 1.00 |
| | AUC@10 | 59.80 | 60.45 | 60.76 | 60.34 $\pm$ 0.49 |
| | AUC@20 | 74.31 | 74.26 | 74.74 | 74.44 $\pm$ 0.27 |
| BlendedMVS | AUC@5 | 72.17 | 73.08 | 73.01 | 72.75 $\pm$ 0.51 |
| | AUC@10 | 83.51 | 84.01 | 83.97 | 83.83 $\pm$ 0.28 |
| | AUC@20 | 90.67 | 90.92 | 90.91 | 90.83 $\pm$ 0.14 |

Table 9: Visual localization results from N=3 independent training runs. We report median translation error (m) / rotation error (deg) ↓.

| Dataset | Split | Run 1 | Run 2 | Run 3 | Mean $\pm$ Std |
|---|---|---|---|---|---|
| 7-Scenes | Average (7) | 0.04 / 1.09 | 0.04 / 1.10 | 0.04 / 1.11 | 0.04$\pm$0.00 / 1.10$\pm$0.01 |
| Cambridge | Average (4)* | 0.37 / 0.53 | 0.38 / 0.55 | 0.38 / 0.52 | 0.38$\pm$0.01 / 0.53$\pm$0.02 |
| Cambridge | Average (5) | 0.51 / 0.52 | 0.52 / 0.53 | 0.52 / 0.51 | 0.52$\pm$0.01 / 0.52$\pm$0.01 |

*Average (4) excludes the GreatCourt scene.

# E ACKNOWLEDGMENTS

We acknowledge the use of LLM assistance for language polishing and grammar correction only in this manuscript.

