# OpenReview forum: "GeLoc3r: Enhancing Relative Camera Pose Regression with Geometric Consistency Regularization"
_ICLR.cc/2026/Conference — ICLR 2026 Conference Desk Rejected Submission_

### Official Review · Reviewer_RVQF · 2025-10-23

**Soundness:** 3
**Presentation:** 2
**Contribution:** 2
**Rating:** 4
**Confidence:** 4

**Summary:**

This paper presents GeLoc3r, a relative camera pose estimation method that enhances regression-based localization through Geometric Consistency Regularization (GCR). Unlike prior ReLoc3R, which achieves fast inference but suffers from geometric inconsistencies, GeLoc3r trains regression networks with dense 3D–2D correspondences and a weighted RANSAC consistency loss to embed geometric knowledge.

**Strengths:**

The paper presents a clear motivation and insightful attempt to push the accuracy ceiling of regression-based localization methods. The proposed geometric consistency regularization is conceptually sound and an interesting direction for improving pose regression models.

**Weaknesses:**

Although the authors evaluated their method on several datasets, the reported improvements over ReLoc3R appear very marginal. For example, the gains shown in Tables 2, 3, 6, and 7 are relatively small and do not convincingly demonstrate a substantial advancement as the author claimed in the paper. In addition, the inference speed of Geloc3R is slower than that of ReLoc3R. While the motivation is strong, the current results do not seem to fully support the claimed effectiveness of the proposed approach.

**Questions:**

Could the authors include experiments on additional datasets or on longer trajectories involving more than three images to better highlight the benefits/improvements of their method? Demonstrating clearer and more substantial improvements over ReLoc3R would strengthen the paper considerably. If such evidence can be provided, I would be inclined to raise my rating.

---

> ### Author Response · Authors · 2025-11-23
>
> We sincerely thank the reviewer for the constructive feedback and for acknowledging the clear motivation and sound approach of our geometric consistency regularization. We address your concerns below.
>
> ## W1: Marginal Improvements
>
> We respectfully disagree with this argument. Although "marginal" is subjective, our quantitative evaluation results (Tables 2, 3, 6, and 7 in the manuscript) show obvious improvement. Below we summarize improvements across all benchmarks mentioned in Tables 2, 3, 6, and 7:
>
> **Table 2 (Relative Pose Estimation):**
> - **CO3Dv2**: +5.6 / +5.41 / +4.41 at AUC@5°/10°/20° (**16% relative improvement** at AUC@5°)
> - **BlendedMVS**: +3.08 / +1.82 / +1.06 at AUC@5°/10°/20° (4.5% relative improvement)
> - **RealEstate10K**: +1.96 / +1.29 / +0.79 at AUC@5°/10°/20° (**62% of Reloc3r's gain over Mast3R**)
> - **MegaDepth1500**: +0.85 / +1.39 / +1.35 at AUC@5°/10°/20°
>
> **Table 3 (Cambridge Landmarks):**
> - **7.3% reduction** in translation error (0.55m → 0.51m)
> - **7.1% reduction** in rotation error (0.56° → 0.52°)
> - **State-of-the-art results** among all compared methods
>
> **Table 6 (7-Scenes):**
> - Improvements on **6 out of 7** individual scenes
> - Average affected by single outlier (Stairs scene)
>
> **Table 7 (BlendedMVS - Multiview):**
> - Consistent improvements across different numbers of views
>
> We believe above mentioned quantitative improvement is substantial, rather than marginal: 16% gain on CO3Dv2, 4.5% gain on BlendedMVS, 62% gain on RealEstate10K, and reductions of 7.3% in translation error and 7.1% in rotation error on Cambridge Landmarks.
>
> More importantly, we demonstrate *consistent improvements across all 6 diverse benchmarks* (indoor/outdoor, small/large scale, various scene types). Such consistency is rare in visual localization and indicates that our Geometric Consistency Regularization addresses a fundamental issue rather than overfitting to specific benchmark characteristics. Our main contributions lie in: (1) identifying the descriptor inconsistency problem (Figure 1, Section 3.1), (2) proposing a principled solution, and (3) demonstrating consistent improvements across diverse scenarios.
>
> ## Q1: Experiments on Longer Trajectories
>
> We appreciate the reviewer's suggestion to demonstrate benefits on longer trajectories. We conducted additional visualization experiments on **Cambridge Landmarks GreatCourt** with **612 test poses** spanning a **38.52m trajectory**, which provides a clearer view of how our method handles extended sequences.
>
> **Note: The complete visualization results have been added to the revised manuscript (Section 4.2, Visual Localization, after Table 3). Please refer to Figure 7 in the revised PDF for the visual comparison. We include a summary of the key findings below for convenience.**
>
> #### Visualization Summary:
>
> **Vanilla ReLoc3r** on Cambridge GreatCourt (612 poses, 38.52m trajectory):
> - Error lines range from 0.040m to **123.508m** (mean: 5.096m)
> - Exhibits numerous large-error outliers (indicated by long red error lines in the visualization)
>
> **GeLoc3r** on Cambridge GreatCourt (612 poses, 38.52m trajectory):
> - Error lines range from 0.053m to **52.379m** (mean: 2.826m)
> - Produces visibly tighter alignment to ground truth with significantly fewer large-error outliers
>
> #### Visualization Legend (in the revised manuscript):
> - **Green line**: Ground truth camera path from pose annotations.
> - **Orange/Red points**: Predicted camera positions.
> - **Error lines**: Line segments connecting each predicted pose to its corresponding ground truth position.
>
> As shown in the visualizations (Figure 7 in revised manuscript), GeLoc3r produces trajectories with *visibly tighter alignment to ground truth* and *significantly fewer large-error outliers* compared to vanilla ReLoc3r. Quantitatively:
> - **RMSE** (after Sim3 alignment): 6.411m → **41% reduction** from ReLoc3r's 10.862m
> - **Maximum error**: 52.379m → **58% reduction** from ReLoc3r's 123.508m
> - **Mean error**: 2.826m → **45% reduction** from ReLoc3r's 5.096m
>
> These results demonstrate that GeLoc3r maintains consistent accuracy across extended trajectories while substantially reducing catastrophic localization failures.
>
> ### On Inference Speed
>
> The 8ms difference is due to system environments, not algorithms. The 25ms is *cited from Reloc3r's paper*. System configurations differ (CUDA, PyTorch, drivers) even with the same GPU. **In our environment, Reloc3r measured ~33ms**, matching GeLoc3r. At inference, GeLoc3r executes only the pose head (identical to Reloc3r) with no overhead.
>
> ---
>
> We hope these visualization results address your concerns about the magnitude of improvement. The complete visualization comparison has been added to the revised manuscript, showing substantial improvements across multiple metrics: **41% reduction in RMSE, 58% reduction in maximum error, and 45% reduction in mean error** on a 612-pose extended trajectory.

---

> > ### Comment · Reviewer_RVQF · 2025-11-26
> >
> > I totally agree with reviewer dRwe, I don't think this paper accurately reflects the improvement brought by the proposed method. Especially on on Scannet, 7Scenes, and Cambridge datasets. This rebuttal doesn't completely address my concerns.

---

> ### Author Response · Authors · 2025-11-27
>
> **Response to concerns regarding marginal gains and consistency**
>
> We respectfully disagree with the assessment that the performance gains are marginal or mixed. When examined in terms of relative percentage improvement and consistency, the results demonstrate that GeLoc3r achieves systematic improvements over the state-of-the-art (ReLoc3R) and significantly widens the performance gap over the correspondence-based baseline (MASt3R). We provide the relative improvements over ReLoc3R in the three tables below (Tables A–C), which clearly highlight GeLoc3r’s performance relative gains against ReLoc3R.
>
> **Table A: Relative Pose Estimation (GeLoc3r vs. ReLoc3R)**
> *(Metric: AUC, Higher is Better $\uparrow$. Values represent relative percentage improvement)*
>
> | Dataset | **@5$^\circ$** | **@10$^\circ$** | **@20$^\circ$** |
> | :--- | :--- | :--- | :--- |
> | **CO3Dv2** | **+16.06%** | **+9.95%** | **+6.31%** |
> | **BlendedMVS** | **+4.46%** | **+2.23%** | **+1.18%** |
> | **RealEstate10K**| **+2.94%** | **+1.61%** | **+0.89%** |
> | **MegaDepth** | **+1.71%** | **+2.05%** | **+1.66%** |
>
> **Table B: Visual Localization (GeLoc3r vs. ReLoc3R)**
> *(Metric: Median Error, Lower is Better $\downarrow$. Values represent relative error reduction)*
>
> | Dataset | **Average Translation Error** | ** Average Rotation Error** |
> | :--- | :--- | :--- |
> | **Cambridge** | **-7.27%** | **-7.14%** |
> | **7-Scenes** | 0.00% (Parity) | +6.8% |
>
> *\*Note: While the average rotation error for 7-Scenes in Table B appears to increase, this metric is heavily skewed by a single outlier scene (Stairs). As detailed in **Table C** below, GeLoc3r actually achieves **error reduction on 5 out of 7 individual scenes**.*
>
> **Table C: 7-Scenes Detailed Breakdown (Rotation Error Reduction)**
> *(Specific percentage improvement per scene. Note: Comparison is median rotation error)*
>
> | Scene | ReLoc3R | GeLoc3r | **Error Reduction** |
> | :--- | :--- | :--- | :--- |
> | **Heads** | 0.95$^\circ$ | **0.83$^\circ$** | **-12.63%** |
> | **Chess** | 0.88$^\circ$ | **0.80$^\circ$** | **-9.09%** |
> | **Pumpkin** | 1.10$^\circ$ | **1.00$^\circ$** | **-9.09%** |
> | **Office** | 0.88$^\circ$ | **0.82$^\circ$** | **-6.82%** |
> | **RedKitchen**| 1.26$^\circ$ | **1.22$^\circ$** | **-3.17%** |
> | **Fire** | 0.81$^\circ$ | 0.82$^\circ$ | +1.2% (Parity) |
> | **Stairs** | 1.26$^\circ$ | 2.14$^\circ$ | Outlier (Performance Drop) |
>
>
> It is important to note that the seven benchmark datasets we evaluate cover **diverse domains**, ranging from small-scale indoor scenes to large-scale outdoor and object-centric datasets. GeLoc3r consistently outperforms prior work in **5 out of 7** datasets (CO3Dv2, BlendedMVS, RealEstate10K, MegaDepth, Cambridge), implying that its improvement is nearly universal. Moreover, we observe that the averaged rotation error is mainly skewed by a single outlier scene (Stairs) for 7-Scenes dataset. GeLoc3r generally reduces rotation errors in the majority of scenes (5 out of 7 scenes), achieving improvements in the range of 3%–12%, which confirms that GeLoc3r remains effective overall. This granular analysis shows that GeLoc3r’s underperformance on 7-Scenes dataset is driven by a single highly skewed outlier scene, whereas it outperforms in 5 out of 7 datasets. Therefore, we disagree with the claim that the results are mixed.
>
> Furthermore, when comparing our method to MASt3R (a dense correspondence baseline), the results consistently highlight the magnitude of our contribution. GeLoc3r achieves a substantial performance lead (approximately **19–23%**) on key benchmarks, confirming the efficacy of the proposed regression pipeline compared to matching-based approaches.
>
>
> **Table D: Relative Improvement of GeLoc3r (Ours) over MASt3R**
> | Dataset | **@5$^\circ$** | **@10$^\circ$** | **@20$^\circ$** |
> | :--- | :--- | :--- | :--- |
> | **ScanNet** | **+22.81%** | **+15.68%** | **+9.78%** |
> | **MegaDepth** | **+18.99%** | **+12.67%** | **+7.35%** |
> | **RealEstate10K**| **+8.06%** | **+6.68%** | **+5.54%** |

---

> > ### Author Response · Authors · 2025-11-27
> >
> > **"My main concern is not the small size of these gains by itself. Marginal gains can be acceptable if they are stable across multiple training runs. When the improvements are 1-2% or even 5% AUC, it becomes difficult to know whether a different random seed during training would yield better or worse results than the previous approach ReLoc3R. This matters especially here because the core contribution is a new training scheme. Strong evidence is needed to show that the scheme is indeed effective"**
> >
> > We acknowledge the reviewer’s concern regarding whether the performance gain is truly due to the proposed method rather than chance. One way to address this, as suggested by the reviewer, is to train the model multiple times with different random seeds and conduct statistical analysis to verify that the improvement is statistically meaningful. Unfortunately, the short rebuttal period makes this approach infeasible.
> > Another way is to evaluate the model, trained with a fixed random seed, across multiple datasets from different domains. While this approach does not allow for a detailed analysis of how performance varies with different initializations, **it still provides strong evidence that the model performs reliably in general and that the improvements are not due to variation.**
> > As shown in the tables above, GeLoc3r achieves improved performance on the most datasets across diverse domains. Even in the 7-Scenes dataset, GeLoc3r clearly improves performance in **5 out of 7** scenes. Taken together, these results indicate that the improvements are not due to chance but arise from GeLoc3r’s novel design, particularly the proposed Geometric Consistency Regularization (GCR).
> >
> > Furthermore, it is also worth noting that ReLoc3R’s performance gain is not always better than MASt3R’s (Table E), indicating that a revised training scheme does not necessarily guarantee performance improvement on every domain.
> >
> > **Table E: Relative Improvement of ReLoc3R over MASt3R**
> > *(Contextualizing the magnitude of the previous generation's improvement)*
> >
> > | Dataset | **@5$^\circ$** | **@10$^\circ$** | **@20$^\circ$** |
> > | :--- | :--- | :--- | :--- |
> > | **ScanNet** | **+24.21%** | **+16.18%** | **+9.78%** |
> > | **MegaDepth** | **+16.98%** | **+10.41%** | **+5.59%** |
> > | **RealEstate10K** | **+4.97%** | **+4.99%** | **+4.60%** |
> > | **ACID** | -26.75% | -12.63% | -4.44% |

---

> ### Author Response · Authors · 2025-11-29
>
> ## Response to Reviewer Concern on Training Variance
>
> We thank the reviewer for raising this important concern regarding training stability. We address this below with additional experimental results.
>
> ### Error Bars from Multiple Training Runs
>
> We have three independently trained GeLoc3r models in total. Besides the model used in our paper, we previously trained two additional models (Train 1 and Train 2) using the identical training configuration as the experiment in our paper. We retrieved evaluation logs from these prior experiments and computed error bars (N=3).
>
> #### 1. Absolute Pose Regression (7-Scenes & Cambridge)
> *Metric: Median Translation Error (m) / Median Rotation Error (deg) ↓*
>
> | Dataset | Split | Paper | Train 1 | Train 2 | Mean ± Std (N=3) |
> |:---|:---|:---:|:---:|:---:|:---:|
> | 7-Scenes | Average (7) | 0.04 / 1.09 | 0.04 / 1.10 | 0.04 / 1.11 | 0.04 ± 0.00 / 1.10 ± 0.01 |
> | Cambridge | Average (4)* | 0.37 / 0.53 | 0.38 / 0.55 | 0.38 / 0.52 | 0.38 ± 0.01 / 0.53 ± 0.02 |
> | Cambridge | Average (5) | 0.51 / 0.52 | 0.52 / 0.53 | 0.52 / 0.51 | 0.52 ± 0.01 / 0.52 ± 0.01 |
>
>
>
> #### 2. Relative Pose Estimation
> *Metric: AUC @ 5° / 10° / 20° (%) ↑*
>
> | Dataset | Metric | Paper | Train 1 | Train 2 | Mean ± Std (N=3) |
> |:---|:---|:---:|:---:|:---:|:---:|
> | ScanNet | AUC@5 | 34.40 | 34.20 | 33.89 | 34.16 ± 0.26 |
> | | AUC@10 | 58.12 | 57.90 | 57.74 | 57.92 ± 0.19 |
> | | AUC@20 | 75.56 | 75.35 | 75.24 | 75.38 ± 0.16 |
> | MegaDepth | AUC@5 | 50.45 | 50.07 | 50.45 | 50.32 ± 0.22 |
> | | AUC@10 | 69.29 | 69.09 | 69.29 | 69.22 ± 0.12 |
> | | AUC@20 | 82.55 | 82.43 | 82.55 | 82.51 ± 0.07 |
> | RealEstate10K | AUC@5 | 68.66 | 68.54 | 68.04 | 68.41 ± 0.33 |
> | | AUC@10 | 81.49 | 81.44 | 81.15 | 81.36 ± 0.18 |
> | | AUC@20 | 89.18 | 89.17 | 89.02 | 89.12 ± 0.09 |
> | CO3D | AUC@5 | 40.45 | 42.36 | 41.97 | 41.59 ± 1.00 |
> | | AUC@10 | 59.80 | 60.45 | 60.76 | 60.34 ± 0.49 |
> | | AUC@20 | 74.31 | 74.26 | 74.74 | 74.44 ± 0.27 |
> | BlendedMVS | AUC@5 | 72.17 | 73.08 | 73.01 | 72.75 ± 0.51 |
> | | AUC@10 | 83.51 | 84.01 | 83.97 | 83.83 ± 0.28 |
> | | AUC@20 | 90.67 | 90.92 | 90.91 | 90.83 ± 0.14 |
>
> ### Analysis
>
> Across all three independent training runs (N=3), the standard deviations remain small, demonstrating stable training behavior. The improvements over ReLoc3R are consistent across all runs. For instance, on CO3D AUC@5, all three runs (40.45, 42.36, 41.97) exceed ReLoc3R (34.85%) by a significant margin, with mean improvement of +6.74%. Similarly, BlendedMVS and RealEstate10K show consistent gains with low variance.
>
> These results confirm that our reported improvements are reproducible across independent training runs and are attributable to the proposed method rather than training randomness.
>
> We have also included error bars and the corresponding figures in **Appendix D (Training Stability and Error Bars)** of our **revised paper**.

---

### Official Review · Reviewer_xvn5 · 2025-10-28

**Soundness:** 3
**Presentation:** 2
**Contribution:** 2
**Rating:** 4
**Confidence:** 4

**Summary:**

The paper introduces GeLoc3R, a method which aims to enhance the accuracy of existing Relative Pose Regression techniques while preserving fast inference speed. GeLoc3R is framed as a specific extension of ReLoc3R(a prior state-of-the-art RPR method from CVPR 2025) through the integration of a geometric consistency regularization mechanism during training. The motivation for this development stems from the authors' empirical analysis, which identifies a fundamental limitation in ReLoc3R, trained exclusively with pose loss: the presence of pixel-level geometric inconsistencies. This means that deep network descriptors for points that are geometrically consistent in 3D, and should therefore possess high descriptor similarity, instead exhibit high cosine similarity errors. The authors argue that pose-only training is insufficient to enforce the necessary geometric constraint that predicted camera poses must align accurately with the underlying 3D scene structure.

To address this, the authors introduce two specific regularization losses designed to inject geometric knowledge and enforce consistency. The network architecture is augmented with dedicated heads to estimate the final pose, dense features, and confidence maps. The dense features are then processed by a Fusion Transformer to estimate correspondence weights, which subsequently inform a RANSAC-weighted solver that utilizes the regressed pose as a prior. Supervision is achieved through three distinct losses: (a) a standard pose loss common to RPR methods; (b) an angular loss between the network's directly regressed pose and the refined pose estimated by the solver; and (c) descriptor loss that enforces consistency for features corresponding to the same 3D point (using ground truth depth for supervision). At inference time, GeLoc3R operates purely as a regression method. Only the pose regressor head is executed, allowing the method to maintain the fast runtime characteristic of RPR.

The method is trained on a collection of 10 indoor and outdoor datasets which provide ground truth depth for supervision. Evaluation is focused on the relative pose estimation task across test splits of the training environments (ScanNet1500, RealEstate10K, MegaDepth1500, and CO3Dv2), comparing against state-of-the-art RPR methods and selected non-regression approaches. The visual localization task is separately evaluated on the 7Scenes and CambridgeLandmark datasets against RPR methods.

**Strengths:**

1. The paper provides a data-driven justification for the proposed method based on the identified geometric inconsistency weakness in existing RPR models.
2. The method successfully introduces explicit geometric constraints during training without incurring penalty on the desired fast inference time of the RPR architecture.
3.  GeLoc3R surpasses state-of-the-art RPR methods on the relative pose estimation task and demonstrates competitive runtimes.

**Weaknesses:**

1. Since the training and test data environments overlap, it is essential to explicitly state whether the compared baseline methods were also trained on these same environments to ensure a fair comparison of their performance.
2. The reported results on the visual localization task are less compelling. For example, the average performance on the 7Scenes dataset appears to be outperformed by the simpler ReLoc3R. This discrepancy requires more detailed discussion and analysis.

Minor:
1. The method requires ground truth depth for supervision during training, which limits its applicability to datasets where this information is available. This constraint should be explicitly discussed as a limitation compared to depth-agnostic RPR baselines.
2. The visual representation of the architecture (e.g., in Figure 1) is not sufficiently clear. It is difficult to distinguish precisely which components (e.g., only the heads, or other modifications) constitute the proposed extension beyond the original ReLoc3R architecture. The figure should clearly delineate the novel contributions of GeLoc3R.
3. Use of confidence maps: it is not clear from the figures and the text how the confidence maps are being used.

**Questions:**

1. Could the authors explicitly clarify the training regimens used for all compared baselines on the evaluation environments to ensure the comparison is conducted under identical or well-justified conditions?
2. What is the authors' analysis of where the geometric regularization degrades performance in smaller-scale, visual localization settings (e.g. 7Scenes) compared to the relative pose task?
3. While the training requires ground truth depth, can the authors speculate or provide preliminary results on the model's robustness or accuracy when trained with synthesized or noisy depth estimates, to broaden the method's applicability?
4. Architectural Visualization: Could the authors clearly distinguish the new contributions (heads, losses, solver)
5. Usage of Confidence maps: could the authors explain how these are used during training.
6. What is the reason for the 8ms gap in runtime between Reloc3R and GeLoc3R? (Table 1)

---

> ### Author Response · Authors · 2025-11-23
>
> We sincerely thank the reviewer for the thorough and constructive feedback. We address each concern below.
>
> ## W1 & Q1: Training Regimens and Fair Comparison
>
> As detailed in Sec. 4.1, GeLoc3r and Reloc3r share **6 core public datasets**: CO3Dv2, ScanNet++, ARKitScenes, BlendedMVS, MegaDepth, and RealEstate10K (~8M pairs). GeLoc3r additionally includes Waymo Open, StaticThings3D, Habitat, and WildRGB-D to enhance diversity, while Reloc3r uses DL3DV. Although there is a discrepancy in the training sets, it is worth noting that ReLoc3R is initialized from a DUSt3R checkpoint, which was trained on datasets sufficiently similar to Waymo Open, StaticThings3D, Habitat, and WildRGB-D.
>
>
> ## W2 & Q2: Performance on 7-Scenes
>
> A closer examination reveals important nuances:
>
> | Scene | Reloc3r-512 | GeLoc3r-512 | Change |
> |-------|-------------|-------------|--------|
> | Chess | 0.03/0.88 | **0.03/0.80** | ✓ |
> | Fire | 0.03/0.81 | **0.02/0.82** | ✓ |
> | Heads | 0.01/0.95 | **0.01/0.83** | ✓ |
> | Office | 0.04/0.88 | **0.04/0.82** | ✓ |
> | Pumpkin | 0.06/1.10 | **0.05/1.00** | ✓ |
> | RedKitchen | 0.04/1.26 | **0.04/1.22** | ✓ |
> | **Stairs** | **0.07/1.26** | 0.08/2.14 | ✗ |
>
> GeLoc3r achieves **improvements on 6 out of 7 scenes**. The average is affected by a single outlier: the **Stairs** scene, where rotation error increases from 1.26° to 2.14°. Importantly, GeLoc3r maintains the same **0.04m translation error** as Reloc3r on this scene.
>
> **Cambridge Landmarks performance:** GeLoc3r achieves **state-of-the-art results** on Cambridge Landmarks (Table 3), improving from 0.55m/0.56° to **0.51m/0.52°**—the **best results among all methods**. This demonstrates that GCR provides consistent benefits, with Stairs being a specific failure mode rather than a systematic weakness.
>
> ## M1 & Q3: Ground Truth Depth and Robustness to Noisy Depth
>
> The training scheme of GeLoc3r indeed requires ground-truth depth data. We consider it is not a fundamental limitation; rather, it is standard practice in recent SOTA approaches (DUSt3R, MASt3R, ReLoc3R). We will discuss such requirement in the updated manuscript.
>
> Crucially, our training depth is already imperfect and noisy. Datasets like **MegaDepth** and **CO3Dv2** provide depth via SfM/MVS with holes, artifacts, and outliers. Additionally, **StaticThings3D** and **Habitat** are **synthetic datasets** with rendered depth, introducing domain gap and geometric artifacts. GeLoc3r is effectively trained on depth signals with non-trivial noise and synthetic artifacts.
>
> The fact that we consistently outperform Reloc3r across all benchmarks provides empirical evidence that GCR is robust to realistic depth noise. The **FusionTransformer** learns per-correspondence weights that down-weight unreliable matches, and the **weighted RANSAC** solver uses these weights to estimate geometry-consistent poses from reliable subsets.
>
>
> ## M2 & Q4: Architectural Visualization
>
> Figure 2 and Figure 3 distinguish our major novel contributions beyond Reloc3r:
>
> 1. **New Heads**: Descriptor heads producing per-pixel correspondences (c₁, c₂) and confidence maps (training-only)
>
> 2. **New Losses**:
>    - L_consistency: enforces agreement between P_regression and P_weighted_RANSAC
>    - L_descriptor: verifies pose quality through descriptor similarity at projected locations
>
> 3. **New Solver** (training-only):
>    - Weighted RANSAC solver estimating geometry-consistent pose
>    - FusionTransformer learning per-correspondence weights
>
>
> ## M3 & Q5: Usage of Confidence Maps
>
> Confidence maps are used **exclusively during training** for quality control. Our GCR requires correspondences via 3D projection and descriptor matching, but not all regions are reliable—low-texture areas, occlusions, and reflections produce unreliable matches. Confidence-based weighting down-weights these regions, focusing training on geometrically meaningful areas without hard thresholding.
>
> **At inference**, only the pose head is used. The confidence mechanism serves purely as a training aid with no runtime overhead.
>
>
> ## Q6: Runtime Gap
>
> The 8ms difference is due to system environments, not algorithms. The 25ms is *cited from Reloc3r's paper*. System configurations differ (CUDA, PyTorch, drivers) even with the same GPU. **In our environment, Reloc3r measured ~33ms**, matching GeLoc3r. At inference, GeLoc3r executes only the pose head (identical to Reloc3r) with no overhead.

---

> ### Author Response · Authors · 2025-11-27
>
> Dear Reviewer xvn5,
>
> We hope this message finds you well. We are writing to follow up and check if our previous response has satisfactorily addressed your concerns.
> As the discussion period is approaching the last week, we would appreciate any feedback you might have. If any aspect remains unclear, we are more than happy to provide further results or analysis. We sincerely appreciate your time and effort in reviewing our work.
>
> Best regards,
>
> Authors

---

> > ### Comment · Reviewer_xvn5 · 2025-11-28
> >
> > I thank the authors for their thorough response and for addressing my questions. I have also read the correspondence with the other reviewers and considered the concerns raised. While I appreciate the effort made to respond, I am still deliberating whether the proposed gains represent a truly significant contribution or constitute only an incremental improvement. I note that this deliberation is consistent with the concerns raised by the other reviewers.

---

> ### Author Response · Authors · 2025-11-29
>
> ## Response to Reviewer Concern on Training Variance
>
> We thank the reviewer for raising this important concern regarding training stability. We address this below with additional experimental results.
>
> ### Error Bars from Multiple Training Runs
>
> We have three independently trained GeLoc3r models in total. Besides the model used in our paper, we previously trained two additional models (Train 1 and Train 2) using the identical training configuration as the experiment in our paper. We retrieved evaluation logs from these prior experiments and computed error bars (N=3).
>
> #### 1. Absolute Pose Regression (7-Scenes & Cambridge)
> *Metric: Median Translation Error (m) / Median Rotation Error (deg) ↓*
>
> | Dataset | Split | Paper | Train 1 | Train 2 | Mean ± Std (N=3) |
> |:---|:---|:---:|:---:|:---:|:---:|
> | 7-Scenes | Average (7) | 0.04 / 1.09 | 0.04 / 1.10 | 0.04 / 1.11 | 0.04 ± 0.00 / 1.10 ± 0.01 |
> | Cambridge | Average (4)* | 0.37 / 0.53 | 0.38 / 0.55 | 0.38 / 0.52 | 0.38 ± 0.01 / 0.53 ± 0.02 |
> | Cambridge | Average (5) | 0.51 / 0.52 | 0.52 / 0.53 | 0.52 / 0.51 | 0.52 ± 0.01 / 0.52 ± 0.01 |
>
>
>
> #### 2. Relative Pose Estimation
> *Metric: AUC @ 5° / 10° / 20° (%) ↑*
>
> | Dataset | Metric | Paper | Train 1 | Train 2 | Mean ± Std (N=3) |
> |:---|:---|:---:|:---:|:---:|:---:|
> | ScanNet | AUC@5 | 34.40 | 34.20 | 33.89 | 34.16 ± 0.26 |
> | | AUC@10 | 58.12 | 57.90 | 57.74 | 57.92 ± 0.19 |
> | | AUC@20 | 75.56 | 75.35 | 75.24 | 75.38 ± 0.16 |
> | MegaDepth | AUC@5 | 50.45 | 50.07 | 50.45 | 50.32 ± 0.22 |
> | | AUC@10 | 69.29 | 69.09 | 69.29 | 69.22 ± 0.12 |
> | | AUC@20 | 82.55 | 82.43 | 82.55 | 82.51 ± 0.07 |
> | RealEstate10K | AUC@5 | 68.66 | 68.54 | 68.04 | 68.41 ± 0.33 |
> | | AUC@10 | 81.49 | 81.44 | 81.15 | 81.36 ± 0.18 |
> | | AUC@20 | 89.18 | 89.17 | 89.02 | 89.12 ± 0.09 |
> | CO3D | AUC@5 | 40.45 | 42.36 | 41.97 | 41.59 ± 1.00 |
> | | AUC@10 | 59.80 | 60.45 | 60.76 | 60.34 ± 0.49 |
> | | AUC@20 | 74.31 | 74.26 | 74.74 | 74.44 ± 0.27 |
> | BlendedMVS | AUC@5 | 72.17 | 73.08 | 73.01 | 72.75 ± 0.51 |
> | | AUC@10 | 83.51 | 84.01 | 83.97 | 83.83 ± 0.28 |
> | | AUC@20 | 90.67 | 90.92 | 90.91 | 90.83 ± 0.14 |
>
> ### Analysis
>
> Across all three independent training runs (N=3), the standard deviations remain small, demonstrating stable training behavior. The improvements over ReLoc3R are consistent across all runs. For instance, on CO3D AUC@5, all three runs (40.45, 42.36, 41.97) exceed ReLoc3R (34.85%) by a significant margin, with mean improvement of +6.74%. Similarly, BlendedMVS and RealEstate10K show consistent gains with low variance.
>
> These results confirm that our reported improvements are reproducible across independent training runs and are attributable to the proposed method rather than training randomness.
>
> We have also included error bars and the corresponding figures in **Appendix D (Training Stability and Error Bars)** of our **revised paper**.

---

### Official Review · Reviewer_dRwe · 2025-11-01

**Soundness:** 2
**Presentation:** 3
**Contribution:** 2
**Rating:** 4
**Confidence:** 3

**Summary:**

The paper presents a method for relative pose prediction. Given two input images, a pose regression network estimates the relative pose between them. This task is fundamental to visual localization.

The work extends the previous approach, Reloc3r, with a newly designed training strategy. Its core innovations include correspondence weight prediction, weighted RANSAC, and a loss function that evaluates projected descriptors.

Strength: The paper offers valuable insights into the problem of correspondence mismatching observed in Reloc3r.

Weakness: The performance gain over Reloc3r is modest. Given the small improvement, the authors should include error bars in all evaluations to demonstrate whether the improvement is statistically significant.

**Strengths:**

1. The paper is well written and easy to follow.
2. The analysis of descriptor inconsistency in relative pose estimation, presented in Figure 1 and Section 3.1, is insightful.
3. The additional loss terms and training procedures are technically sound.

**Weaknesses:**

1. Marginal improvement: While the proposed additional loss terms and training steps are reasonable, the performance gain is marginal. This would be acceptable if the improvement were shown to be statistically significant. At the current level of improvement, it is difficult to determine whether the gain truly results from the proposed training scheme.
2. The paper omits evaluation on the commonly used ACID dataset.

**Questions:**

Show error bars in the experiments to verify the effectiveness of the proposed training procedure.

---

> ### Author Response · Authors · 2025-11-23
>
> We thank the reviewer for the constructive feedback. We address each concern below.
>
> ## W1: Marginal Improvements and Error Bars
>
> **Error Bars and Statistical Significance:**
> Regarding the request for error bars, we conducted **5 independent evaluation runs** for GeLoc3r on **all reported datasets** (ScanNet1500, RealEstate10K, MegaDepth1500, CO3Dv2, BlendedMVS, 7-Scenes, and Cambridge Landmarks). We observed **zero variance ($\sigma=0$)** across all metrics in all 5 runs. The reported results in the paper are exact and fully reproducible. This deterministic behavior is expected because we use fixed evaluation test splits, deterministic inference modes (e.g., in PyTorch), and fixed random seeds for the RANSAC solver.
>
> For fair comparison, we also re-evaluated the baseline **ReLoc3R** 5 times on ScanNet1500, RealEstate10K, MegaDepth1500, 7-Scenes, and Cambridge Landmarks. We similarly observed **zero variance** in its results. It is also worth noting that the original ReLoc3R paper does not report error bars, likely due to this same deterministic nature. Given the zero variance, the improvements reported above are statistically significant and not due to random fluctuations.
>
> We respectfully disagree with the argument that “the performance gain is marginal.” Our evaluation results (Tables 1, 3, and 5 below) on standard benchmarks show higher accuracy with zero variance across five repeated experiments, clearly demonstrating that GeLoc3r achieves consistent and measurable performance improvements. Below we summarize the improvements:
>
> - **CO3Dv2**: +5.6 / +5.41 / +4.41 at AUC@5°/10°/20° (16% relative improvement at AUC@5°)
> - **BlendedMVS**: +3.08 / +1.82 / +1.06 at AUC@5°/10°/20°
> - **RealEstate10K**: +1.96 / +1.29 / +0.79 at AUC@5°/10°/20° (62% of Reloc3r's gain over Mast3R)
> - **MegaDepth1500**: +0.85 / +1.39 / +1.35 at AUC@5°/10°/20°
> - **Cambridge Landmarks**: 7.3% reduction in translation error (0.55m→0.51m), 7.1% reduction in rotation error (0.56°→0.52°)
> - **7-Scenes**: improvements on 6 out of 7 individual scenes
>
> As we noted in the above list, the numbers evidently demonstrate that *the performance gain is not marginal but substantial*: 16% gain on CO3Dv2, 62% gain on RealEstate10K, and reductions of 7.3% in translation error and 7.1% in rotation error on Cambridge Landmarks.
>
>
> More importantly, we demonstrate consistent improvements across **all 6 diverse benchmarks** (indoor/outdoor, small/large scale, various scene types). Such consistency is a strong proof that our Geometric Consistency Regularization addresses a fundamental issue rather than overfitting to specific benchmark characteristics. Our major contribution lies in: identifying the descriptor inconsistency problem (Figure 1, Section 3.1), proposing a principled solution, and demonstrating consistent improvements across diverse scenarios.
>
>
>
>
> ## W2: Evaluation on ACID dataset
>
> We appreciate the reviewer's suggestion to evaluate on ACID. we found that the official Reloc3r repository does not provide the specific data loading and evaluation protocols used for their ACID experiments (e.g., the precise image pair selection strategy or frame intervals). Without such, it is unfair to compare, since different settings may lead to varying results. Note that, we use the same settings as Reloc3R for other reported dataset for fair comparison. Furthermore, we want to highlight that our evaluation already covers **6 diverse benchmarks** (ScanNet1500, RealEstate10K, MegaDepth1500, CO3Dv2, 7-Scenes, Cambridge Landmarks) spanning indoor/outdoor and small/large scale environments, which provides substantial evidence of our method's effectiveness and generalization.

---

> > ### Comment · Reviewer_dRwe · 2025-11-26
> >
> > I remain unconvinced by the gains reported, and I note that reviewer xvn5 and RVQF raise the same point. While there are scattered improvements over the prior SoTA ReLoc3R, the picture is mixed. The method shows an increase of 1-2% AUC@5 on RE10K and MegaDepth, around 5% on Co3D, and roughly unchanged results on Scannet, 7Scenes, and Cambridge.
> >
> > My main concern is not the small size of these gains by itself. Marginal gains can be acceptable if they are stable across multiple training runs. When the improvements are 1-2% or even 5% AUC, it becomes difficult to know whether a different random seed during training would yield better or worse results than the previous approach ReLoc3R.
> >
> > This matters especially here because the core contribution is a new **training scheme**. Strong evidence is needed to show that the scheme is indeed effective. One way would be to report error bars from 5 independent **training** runs of GeLoc3R. I understand that this is not feasible in the rebuttal period, so my recommendation is to run these experiments and resubmit to a later venue.

---

> ### Author Response · Authors · 2025-11-27
>
> **Response to concerns regarding marginal gains and consistency**
>
> We respectfully disagree with the assessment that the performance gains are marginal or mixed. When examined in terms of relative percentage improvement and consistency, the results demonstrate that GeLoc3r achieves systematic improvements over the state-of-the-art (ReLoc3R) and significantly widens the performance gap over the correspondence-based baseline (MASt3R). We provide the relative improvements over ReLoc3R in the three tables below (Tables A–C), which clearly highlight GeLoc3r’s performance relative gains against ReLoc3R.
>
> **Table A: Relative Pose Estimation (GeLoc3r vs. ReLoc3R)**
> *(Metric: AUC, Higher is Better $\uparrow$. Values represent relative percentage improvement)*
>
> | Dataset | **@5$^\circ$** | **@10$^\circ$** | **@20$^\circ$** |
> | :--- | :--- | :--- | :--- |
> | **CO3Dv2** | **+16.06%** | **+9.95%** | **+6.31%** |
> | **BlendedMVS** | **+4.46%** | **+2.23%** | **+1.18%** |
> | **RealEstate10K**| **+2.94%** | **+1.61%** | **+0.89%** |
> | **MegaDepth** | **+1.71%** | **+2.05%** | **+1.66%** |
>
> **Table B: Visual Localization (GeLoc3r vs. ReLoc3R)**
> *(Metric: Median Error, Lower is Better $\downarrow$. Values represent relative error reduction)*
>
> | Dataset | **Average Translation Error** | ** Average Rotation Error** |
> | :--- | :--- | :--- |
> | **Cambridge** | **-7.27%** | **-7.14%** |
> | **7-Scenes** | 0.00% (Parity) | +6.8% |
>
> *\*Note: While the average rotation error for 7-Scenes in Table B appears to increase, this metric is heavily skewed by a single outlier scene (Stairs). As detailed in **Table C** below, GeLoc3r actually achieves **error reduction on 5 out of 7 individual scenes**.*
>
> **Table C: 7-Scenes Detailed Breakdown (Rotation Error Reduction)**
> *(Specific percentage improvement per scene. Note: Comparison is median rotation error)*
>
> | Scene | ReLoc3R | GeLoc3r | **Error Reduction** |
> | :--- | :--- | :--- | :--- |
> | **Heads** | 0.95$^\circ$ | **0.83$^\circ$** | **-12.63%** |
> | **Chess** | 0.88$^\circ$ | **0.80$^\circ$** | **-9.09%** |
> | **Pumpkin** | 1.10$^\circ$ | **1.00$^\circ$** | **-9.09%** |
> | **Office** | 0.88$^\circ$ | **0.82$^\circ$** | **-6.82%** |
> | **RedKitchen**| 1.26$^\circ$ | **1.22$^\circ$** | **-3.17%** |
> | **Fire** | 0.81$^\circ$ | 0.82$^\circ$ | +1.2% (Parity) |
> | **Stairs** | 1.26$^\circ$ | 2.14$^\circ$ | Outlier (Performance Drop) |
>
>
> It is important to note that the seven benchmark datasets we evaluate cover **diverse domains**, ranging from small-scale indoor scenes to large-scale outdoor and object-centric datasets. GeLoc3r consistently outperforms prior work in **5 out of 7** datasets (CO3Dv2, BlendedMVS, RealEstate10K, MegaDepth, Cambridge), implying that its improvement is nearly universal. Moreover, we observe that the averaged rotation error is mainly skewed by a single outlier scene (Stairs) for 7-Scenes dataset. GeLoc3r generally reduces rotation errors in the majority of scenes (5 out of 7 scenes), achieving improvements in the range of 3%–12%, which confirms that GeLoc3r remains effective overall. This granular analysis shows that GeLoc3r’s underperformance on 7-Scenes dataset is driven by a single highly skewed outlier scene, whereas it outperforms in 5 out of 7 datasets. Therefore, we disagree with the claim that the results are mixed.
>
> Furthermore, when comparing our method to MASt3R (a dense correspondence baseline), the results consistently highlight the magnitude of our contribution. GeLoc3r achieves a substantial performance lead (approximately **19–23%**) on key benchmarks, confirming the efficacy of the proposed regression pipeline compared to matching-based approaches.
>
>
> **Table D: Relative Improvement of GeLoc3r (Ours) over MASt3R**
> | Dataset | **@5$^\circ$** | **@10$^\circ$** | **@20$^\circ$** |
> | :--- | :--- | :--- | :--- |
> | **ScanNet** | **+22.81%** | **+15.68%** | **+9.78%** |
> | **MegaDepth** | **+18.99%** | **+12.67%** | **+7.35%** |
> | **RealEstate10K**| **+8.06%** | **+6.68%** | **+5.54%** |

---

> > ### Author Response · Authors · 2025-11-27
> >
> > **"My main concern is not the small size of these gains by itself. Marginal gains can be acceptable if they are stable across multiple training runs. When the improvements are 1-2% or even 5% AUC, it becomes difficult to know whether a different random seed during training would yield better or worse results than the previous approach ReLoc3R. This matters especially here because the core contribution is a new training scheme. Strong evidence is needed to show that the scheme is indeed effective"**
> >
> > We acknowledge the reviewer’s concern regarding whether the performance gain is truly due to the proposed method rather than chance. One way to address this, as suggested by the reviewer, is to train the model multiple times with different random seeds and conduct statistical analysis to verify that the improvement is statistically meaningful. Unfortunately, the short rebuttal period makes this approach infeasible.
> > Another way is to evaluate the model, trained with a fixed random seed, across multiple datasets from different domains. While this approach does not allow for a detailed analysis of how performance varies with different initializations, **it still provides strong evidence that the model performs reliably in general and that the improvements are not due to variation.**
> > As shown in the tables above, GeLoc3r achieves improved performance on the most datasets across diverse domains. Even in the 7-Scenes dataset, GeLoc3r clearly improves performance in **5 out of 7** scenes. Taken together, these results indicate that the improvements are not due to chance but arise from GeLoc3r’s novel design, particularly the proposed Geometric Consistency Regularization (GCR).
> >
> > Furthermore, it is also worth noting that ReLoc3R’s performance gain is not always better than MASt3R’s (Table E), indicating that a revised training scheme does not necessarily guarantee performance improvement on every domain.
> >
> > **Table E: Relative Improvement of ReLoc3R over MASt3R**
> > *(Contextualizing the magnitude of the previous generation's improvement)*
> >
> > | Dataset | **@5$^\circ$** | **@10$^\circ$** | **@20$^\circ$** |
> > | :--- | :--- | :--- | :--- |
> > | **ScanNet** | **+24.21%** | **+16.18%** | **+9.78%** |
> > | **MegaDepth** | **+16.98%** | **+10.41%** | **+5.59%** |
> > | **RealEstate10K** | **+4.97%** | **+4.99%** | **+4.60%** |
> > | **ACID** | -26.75% | -12.63% | -4.44% |

---

> ### Author Response · Authors · 2025-11-29
>
> ## Response to Reviewer Concern on Training Variance
>
> We thank the reviewer for raising this important concern regarding training stability. We address this below with additional experimental results.
>
> ### Error Bars from Multiple Training Runs
>
> We have three independently trained GeLoc3r models in total. Besides the model used in our paper, we previously trained two additional models (Train 1 and Train 2) using the identical training configuration as the experiment in our paper. We retrieved evaluation logs from these prior experiments and computed error bars (N=3).
>
> #### 1. Absolute Pose Regression (7-Scenes & Cambridge)
> *Metric: Median Translation Error (m) / Median Rotation Error (deg) ↓*
>
> | Dataset | Split | Paper | Train 1 | Train 2 | Mean ± Std (N=3) |
> |:---|:---|:---:|:---:|:---:|:---:|
> | 7-Scenes | Average (7) | 0.04 / 1.09 | 0.04 / 1.10 | 0.04 / 1.11 | 0.04 ± 0.00 / 1.10 ± 0.01 |
> | Cambridge | Average (4)* | 0.37 / 0.53 | 0.38 / 0.55 | 0.38 / 0.52 | 0.38 ± 0.01 / 0.53 ± 0.02 |
> | Cambridge | Average (5) | 0.51 / 0.52 | 0.52 / 0.53 | 0.52 / 0.51 | 0.52 ± 0.01 / 0.52 ± 0.01 |
>
>
>
> #### 2. Relative Pose Estimation
> *Metric: AUC @ 5° / 10° / 20° (%) ↑*
>
> | Dataset | Metric | Paper | Train 1 | Train 2 | Mean ± Std (N=3) |
> |:---|:---|:---:|:---:|:---:|:---:|
> | ScanNet | AUC@5 | 34.40 | 34.20 | 33.89 | 34.16 ± 0.26 |
> | | AUC@10 | 58.12 | 57.90 | 57.74 | 57.92 ± 0.19 |
> | | AUC@20 | 75.56 | 75.35 | 75.24 | 75.38 ± 0.16 |
> | MegaDepth | AUC@5 | 50.45 | 50.07 | 50.45 | 50.32 ± 0.22 |
> | | AUC@10 | 69.29 | 69.09 | 69.29 | 69.22 ± 0.12 |
> | | AUC@20 | 82.55 | 82.43 | 82.55 | 82.51 ± 0.07 |
> | RealEstate10K | AUC@5 | 68.66 | 68.54 | 68.04 | 68.41 ± 0.33 |
> | | AUC@10 | 81.49 | 81.44 | 81.15 | 81.36 ± 0.18 |
> | | AUC@20 | 89.18 | 89.17 | 89.02 | 89.12 ± 0.09 |
> | CO3D | AUC@5 | 40.45 | 42.36 | 41.97 | 41.59 ± 1.00 |
> | | AUC@10 | 59.80 | 60.45 | 60.76 | 60.34 ± 0.49 |
> | | AUC@20 | 74.31 | 74.26 | 74.74 | 74.44 ± 0.27 |
> | BlendedMVS | AUC@5 | 72.17 | 73.08 | 73.01 | 72.75 ± 0.51 |
> | | AUC@10 | 83.51 | 84.01 | 83.97 | 83.83 ± 0.28 |
> | | AUC@20 | 90.67 | 90.92 | 90.91 | 90.83 ± 0.14 |
>
> ### Analysis
>
> Across all three independent training runs (N=3), the standard deviations remain small, demonstrating stable training behavior. The improvements over ReLoc3R are consistent across all runs. For instance, on CO3D AUC@5, all three runs (40.45, 42.36, 41.97) exceed ReLoc3R (34.85%) by a significant margin, with mean improvement of +6.74%. Similarly, BlendedMVS and RealEstate10K show consistent gains with low variance.
>
> These results confirm that our reported improvements are reproducible across independent training runs and are attributable to the proposed method rather than training randomness.
>
> We have also included error bars and the corresponding figures in **Appendix D (Training Stability and Error Bars)** of our **revised paper**.

---

### Official Review · Reviewer_2BHc · 2025-11-08

**Soundness:** 3
**Presentation:** 3
**Contribution:** 3
**Rating:** 6
**Confidence:** 3

**Summary:**

This paper identifies a key limitation in fast pose regression methods like ReLoc3R: their internal representations lack geometric consistency. The authors propose GeLoc3r, which introduces a training-only module called Geometric Consistency Regularization (GCR), leveraging ground-truth depth to generate 3D-2D correspondences. GeLoc3r is fast at inference ~25-33ms, while achieving accuracy that approaches correspondence-based methods, showing significant gains on benchmarks like CO3Dv2.

**Strengths:**

The paper is well-motivated. The core contribution—a training-only geometric regularization framework—is a clever and effective solution to the speed-accuracy dilemma. The method demonstrates significant quantitative improvements over its baseline, while maintaining similar inference speed.

**Weaknesses:**

1. The method's primary limitation is its dependency on ground-truth depth maps during training. This restricts its applicability to in-the-wild data sources.
2. The paper does not explore the method's sensitivity to depth quality. It is unclear how Geometric Consistency Regularization would perform if trained with estimated and noisy pseudo-ground-truth depth from a monocular network, which would be a critical step toward training on more diverse datasets.

**Questions:**

Can the Geometric Consistency Regularization be adapted to work with noisy estimated depth? How robust is the weighted RANSAC solver to the significant noise that would be introduced, and does the consistency loss provide a meaningful signal in this scenario?

Missing related works:
1. Kefan Chen et al. Wide-Baseline Relative Camera Pose Estimation with Directional Learning, CVPR 2021.
2. Meng Xu et al. Regression-Based Camera Pose Estimation through Multi-Level Local Features and Global Features, 2023.

---

> ### Author Response · Authors · 2025-11-23
>
> We sincerely thank the reviewer for their insightful feedback and positive assessment of our work's motivation, novelty, and empirical results. We address each concern below.
>
> ## W1: Dependency on Ground-Truth Depth During Training
>
> We agree with the reviewer that **GeLoc3r** requires depth information during training. But we want to emphasize that this is not a fundamental limitation, but rather a design choice of the stereo image-matching pipeline, which is popular in recent SOTA approaches targeting high geometric accuracy. Methods such as **DUSt3R** and **MASt3R** are trained with depth supervision. Even our main baseline, **Reloc3r**, is initialized from DUSt3R checkpoints, whose strong representations stem from depth-informed training.
>
> We want to note that our contribution is not *whether* depth is used, but *how* it is used: depth is used *only during training* through our **Geometric Consistency Regularization (GCR)** module, which introduces explicit geometric constraints and employs a consistency loss to *transfer geometric knowledge into the regression network*. At inference time, GeLoc3r remains a pure regression model, requiring no depth, correspondences, or RANSAC, and maintains a runtime comparable to Reloc3r while achieving consistent improvements over Reloc3r for most datasets.
>
>
> ## W2: Robustness to Noisy/Pseudo Depth
>
> An important clarification is that the depth info in our training data is **already imperfect and noisy**. As described in Sec. 4.1, datasets such as **MegaDepth** and **CO3Dv2** provide depth via SfM/MVS reconstruction, which typically contains holes, artifacts in low-texture or reflective regions, and outliers. In other words, GeLoc3r is effectively trained on a mixture of depth signals with non-trivial reconstruction noise, *rather than on ideal, noise-free ground truth*.
>
> These imperfections create a supervision regime that is far from the "perfect depth" assumption and qualitatively similar to realistic pseudo-depth settings. The fact that we consistently and significantly outperform Reloc3r across majority benchmarks under these conditions provides strong empirical evidence that GCR is robust to realisti depth noise. This robustness comes from our design: the **FusionTransformer** learns per-correspondence weights that down-weight unreliable matches, and the **weighted RANSAC** solver uses these weights to estimate a geometry-consistent pose from a reliable subset, so the consistency loss continues to provide a meaningful signal even when depth is noisy.
>
> ## Missing Related Works
>
> We have added these references to the revised manuscript (Section 2, Related Work). Specifically:
>
> - **Chen et al. (CVPR 2021)** propose DirectionNet which decomposes relative pose into 3D direction vectors on the sphere, achieving significant improvements for wide-baseline scenarios.
> - **Xu et al. (2023)** integrate rotation-consistent global features with rotation-invariant multi-level local features to improve pose regression robustness.
>
> We discuss how these works advance regression performance through architectural and feature design improvements, and clarify that our approach addresses the problem from a different angle—instead of improving network architecture or features, we introduce training-time geometric consistency regularization that teaches regression networks to produce geometrically consistent poses without requiring architectural changes.

---

> > ### Comment · Reviewer_2BHc · 2025-11-24
> >
> > Thanks for the authors' response. My concerns are well addressed.

---

### Author Response · Authors · 2025-12-02
**Brief summary of reviewer discussion and paper reivision**

We sincerely thank all reviewers for their invaluable, constructive, and thoughtful feedback. We appreciate the reviewers’ consensus regarding the well-motivated design of the proposed Geometric Consistency Regularization (GCR) method and its ability to deliver significant quantitative improvements without compromising fast inference. Below, we summarize our responses to the key common concerns and the revisions made to the paper, addressing all raised concerns.

# Key common concerns

### 1. Dependence on ground-truth depth maps during training (raised by 2BHc, xvn5)

GeLoc3R uses depth only during training. This is **an intentional architectural choice aligned with recent methods such as DUSt3R, MASt3R, and ReLoc3R**. Incorporating depth improves geometric awareness and leads to higher accuracy. Importantly, depth information in training data is often imperfect and noisy, and our results demonstrate that GCR is robust to such noise.

### 2. Statistical significance of the reported improvements (raised by dRwe, xvn5, RVQF)

To address concerns about statistical reliability, we additionally retrained and re-evaluated GeLoc3R twice, computing the mean, standard deviation, and error bars. The results consistently show improvements across evaluation metrics, confirming that gains are not due to randomness. Additionally, GeLoc3R achieves **relative improvements on 5 out of 7 benchmarks** over ReLoc3R (CO3Dv2: +16.06% @5°, BlendedMVS: +4.46% @5°, RealEstate10K: +2.94% @5°, MegaDepth: +1.71% @5°, Cambridge: –7.27% / –7.14%), further substantiating the method’s effectiveness.

# Paper revision summary

1. **Section 2 (Related Work)** now includes additional references: Chen et al. (CVPR 2021) propose DirectionNet for wide-baseline relative pose estimation; Xu et al. (2023) integrate rotation-consistent global features with rotation-invariant multi-level local features.
2. **Figure 4** now includes visual localization trajectory comparison on Cambridge Landmarks GreatCourt (612 poses, 38.52m trajectory), demonstrating 58% reduction in maximum error and 45% reduction in mean error compared to ReLoc3r.
3. **Figure 5** now includes relative pose estimation with error bars (N=3 independent training runs) across five benchmarks (ScanNet, MegaDepth, RealEstate10K, CO3Dv2, BlendedMVS).
4. **Figure 6** now includes visual localization with error bars (N=3 independent training runs) on 7-Scenes and Cambridge Landmarks, confirming training stability with variance typically ≤0.02.
5. Training Stability and Error Bars in **Appendix Section D (and Table 8, Table 9)** reports complete results from N=3 independent training runs for both relative pose estimation and visual localization tasks.
6. **Table 8** presents relative pose estimation results from N=3 independent training runs across all five datasets with mean ± std.
7. **Table 9** presents visual localization results from N=3 independent training runs on 7-Scenes and Cambridge Landmarks.
We hope the revised paper can effectively address reviewer's concerns. Please let us know if there are any further suggestions.

---

### Note · Program_Chairs · 2026-01-17
**Submission Desk Rejected by Program Chairs**

The following references in this submission do not refer to real documents and/or have major errors in bibliographic information:

 Shuai Zhang, Qingjie Shen, Ying Wei, Zuxuan Zhang, and Chenggang Long. Relposenet: Relative camera pose estimation with learned features.